# Beyond English-Centric Training: How Reinforcement Learning Improves Cross-Lingual Reasoning in LLMs

**Shulin Huang**[1,2]*, **Yiran Ding**[2]*, **Junshu Pan**[1,2]*, **Yue Zhang**[2]†

[1]Zhejiang University, [2]Westlake University

{huangshulin,dingyiran,panjunshu,zhangyue}@westlake.edu.cn

## Abstract

Enhancing the complex reasoning capabilities of Large Language Models (LLMs) attracts widespread attention. While reinforcement learning (RL) has shown superior performance for improving complex reasoning, its impact on cross-lingual generalization compared to Supervised Fine-Tuning (SFT) remains unexplored. We present the first systematic investigation into cross-lingual reasoning generalization of RL and SFT. Using Qwen2.5-3B-Base as our foundation model, we conduct experiments on diverse multilingual reasoning benchmarks, including math reasoning, commonsense reasoning, and scientific reasoning. Our investigation yields two significant findings: (1) Tuning with RL not only achieves higher accuracy but also demonstrates substantially stronger cross-lingual generalization capabilities compared to SFT. (2) RL training on non-English data yields better overall performance and generalization than training on English data, which is not observed with SFT. Furthermore, through comprehensive mechanistic analyses, we explore the underlying factors of RL's superiority and generalization across languages. Our results provide compelling evidence that RL enables the model with more robust reasoning strategies, offering crucial guidance for more equitable and effective multilingual reasoning.

## 1 Introduction

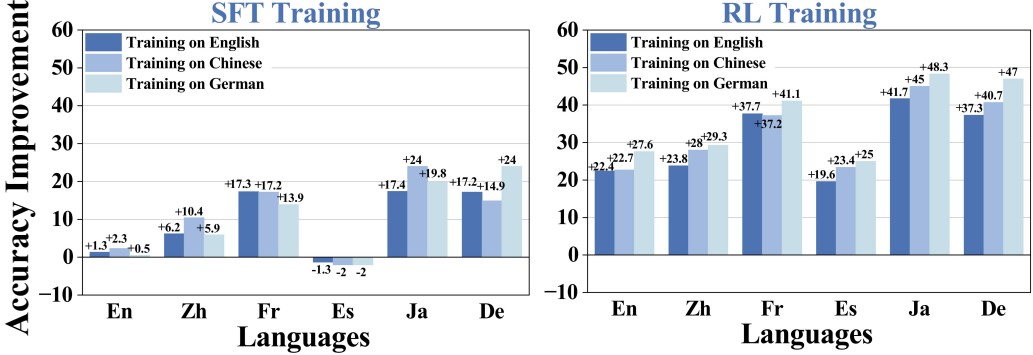

Figure 1: SFT and RL performance improvement using English, Chinese and German training data on the same base model, Qwen2.5-3B-Base. Performance improvements are measured relative to the base model. We report the performance improvement in six language settings.

Multilingualism plays a significant role in human society and occupies a critical position in the development of large language models (LLMs). With over 7,000 languages worldwide, each encapsulates unique cultural contexts and expressive modalities (Campbell & Grondona, 2008). LLMs

---

* indicates equal contribution.

† Correspondence to: (zhangyue@westlake.edu.cn)

not only break down language barriers and facilitate cross-cultural communication, but also enable equitable global Artificial Intelligence benefits (Howard & Ruder, 2018; Sharma et al., 2025).

With advances in LLMs reasoning (Hao et al., 2023; Yue et al., 2025), cross-lingual reasoning has received increasing attention (Wang et al., 2024a; Chai et al., 2025; Payoungkhamdee et al., 2025). Multilingual reasoning requires models to not only comprehend the semantic content of different languages, but also to possess the ability to perform logical inference and problem-solving across diverse linguistic environments (Alam et al., 2024). Current research demonstrates that while large-scale pre-trained language models have achieved remarkable progress in English comprehension and generation, significant performance gaps persist for other languages (Qiu et al., 2024). Consequently, enhancing multilingual reasoning capabilities and achieving effective cross-lingual generalization has emerged as a significant challenge (Pham et al., 2024; Hu et al., 2025).

Reinforcement Learning (RL) is considered a pivotal tool for enhancing reasoning capabilities (Wang et al., 2024b; Guo et al., 2025). Recent studies show that RL training exhibits superior performance on complex tasks such as mathematical and logical reasoning (Xie et al., 2025). Compared to Supervised Fine-tuning (SFT), Reinforcement Learning, through reward-guided mechanisms, enables models with more robust and generalizable reasoning strategies (Huan et al., 2025). Notably, researches reveal that RL not only significantly improves model performance, but also enables stronger *cross-task* generalization capabilities (Shao et al., 2024). In this work, we conduct the first investigation into whether RL exhibits strong generalization capabilities across languages in reasoning. Experimentally, we compare the performance of RL and SFT on diverse languages, exploring their performance across various languages to examine their *cross-lingual* generalization abilities. To fully reflect reasoning capability, we evaluate performance on diverse multilingual reasoning benchmarks, including math reasoning, commonsense reasoning, and scientific reasoning (Shi et al.; She et al., 2024; Son et al., 2025; Xuan et al., 2025; Qi et al., 2025).

The empirical investigation yields two notable findings: (1) As illustrated in Figure 1, RL demonstrates superior performance improvements compared to SFT, with enhanced cross-lingual generalization capabilities. Our results indicate that models trained with RL can more effectively transfer reasoning abilities learned in one language to another. This finding of cross-lingual reasoning is consistent with existing findings for cross-task transfer (Korkmaz, 2024; Huan et al., 2025; Cheng et al., 2025; Chu et al.). (2) Given that the pre-training corpora of most existing LLMs are predominantly English-centric (Morishita et al., 2024; Rytting & Wingate, 2021; Singh et al., 2024), the conventional expectation is that RL training with English data would maximally leverage the model's potential (Yoon et al., 2024; She et al., 2024). However, as shown in Figure 1, our findings surprisingly reveal a counter-intuitive phenomenon: RL training using non-English data (such as Chinese and German) yields better cross-lingual reasoning performance and superior generalization than using English data. This contrasts with SFT, where no such phenomenon is observed, as performance remains comparable and, in some datasets, even shows an opposite trend.

To investigate underlying reasons for the findings, we conduct preliminary analyses: (1) First, we analyze whether the language used in reasoning is consistent with the language of input question. Our investigation reveals that language inconsistency serves as a potential factor of RL's cross-lingual generalization, and the superiority of the non-English data in RL. (2) Second, we examine the role of sampling mechanisms in RL's superior performance. We find that the sampling mechanism in RL explores sufficient and diverse solution paths, allowing models to learn more robust and generalizable strategies. (3) Third, we explore the semantic shift of the model after training. We find that the stability of the semantic space contributes to RL's superior cross-lingual generalization. Our preliminary explorations provide insights for future research in multilingual reasoning.

The main contributions of this work are as follows:

(1) We present the first systematic analysis of the differences between RL and SFT in cross-lingual reasoning generalization, filling an important gap in this research area.

(2) We reveal two significant findings: 1) RL excels over SFT in cross-lingual generalization, and 2) Counterintuitively, non-English data is superior to English data for RL training. To our knowledge, we are the first to demonstrate that using non-English data for RL more effectively enhances performance and cross-lingual generalization, although most models are pre-trained mainly on English.

(3) Through comprehensive analyses, we explore three potential factors underlying RL's superiority: linguistic inconsistency in reasoning, sampling-driven policy optimization, and the semantic shift after training, which provides a crucial foundation for multilingual reasoning.

## 2 RELATED WORK

**Multilingual Reasoning.** Multilingual reasoning is a challenging and representative task for evaluating the intelligence of large language models (Ahn et al., 2024; She et al., 2024; Yoon et al., 2024; Chen et al., 2024). Shi et al. establish the foundation for this field by translating English mathematical problems from GSM8K (Cobbe et al., 2021) into multiple languages, creating the multilingual benchmark MGSM (Shi et al.). To enhance multilingual reasoning capabilities, existing work primarily employs prompting strategies. Qin et al. (2023) and Huang et al. (2023) propose a translate-then-solve approach that first translates non-English questions into English before problem-solving, achieving promising results on closed-source models like ChatGPT (Ouyang et al., 2022). However, less attention has been paid to how different training paradigms affect the model's intrinsic cross-lingual generalization capabilities. Our work addresses this gap by comparing SFT and RL at the model's foundational level, demonstrating RL's unique advantage in learning reasoning strategies without relying on specific languages.

**Supervised Fine-tuning For Reasoning.** Supervised fine-tuning (SFT) effectively enhances LLM reasoning abilities by distilling expert-level chain-of-thought (CoT) examples (Huang et al., 2024). Synthetic data generation is a key strategy: Large teacher models are used to generate solutions for mathematical problems (Yue et al., 2023; Tang et al., 2024), enhancing the reasoning process. Additionally, recent research examines the impact of data quality factors on the model performance (Toshniwal et al., 2025; Yu et al., 2023; Ye et al., 2025). Beyond mathematics, other works (Kim et al., 2023; Xu et al., 2024) expand reasoning tasks to larger domains, broadening the scope and complexity of problem-solving in various fields. Although SFT is successful in enhancing reasoning, its learning approach is fundamentally based on imitating and memorizing given "expert" trajectories (Ge et al., 2023), leading to overfitting to the language and pattern in the training data. Our research differs from these works by focusing on the limitations of SFT in cross-lingual scenarios. By contrasting it with RL, we demonstrate that merely imitating high-quality CoT data is insufficient for achieving the robust cross-lingual generalization that RL provides.

**Reinforcement Learning For Reasoning.** Reinforcement learning (RL) has become a widely adopted technique for post-training large language models (LLMs) to better align their outputs with human preferences (Ouyang et al., 2022; Achiam et al., 2023). Recent studies extend its application to enhancing reasoning abilities, encouraging longer, structured CoT traces and occasional breakthrough moments (Jaech et al., 2024; Guo et al., 2025). These approaches treat chain-of-thought (CoT) reasoning as an RL problem, utilizing various reward mechanisms such as final-answer correctness (Xie et al., 2025; Wen et al., 2025), verifier-based scoring (Gehring et al., 2025), and step-level rewards (Zhang et al., 2025). While online RL approaches (Schulman et al., 2017; Shao et al., 2024) are commonly used, high computational costs motivate the development of offline RL methods (Zhang et al., 2024; Yuan et al., 2025). Existing work primarily relies on English-centric data in RL. In this work, we not only validate the effectiveness of RL in a multilingual setting but also innovatively uncover the unique advantages of non-English training data within the RL framework.

## 3 RL IMPROVES THE GENERALIZATION ACROSS LANGUAGES

### 3.1 EXPERIMENTAL SETUP

**Base Model and Datasets.** We adopt Qwen2.5-3B-Base (Yang et al., 2024) as the base model to clearly explore the impact of RL and SFT. To further examine the generality of the observed phenomena, we also include SmolLM3-3B-Base (Bakouch et al., 2025) and Qwen2.5-7B-Base as the additional base models for verification. The training datasets are translations of GSM8K (Cobbe et al., 2021) and LUFFY (Yan et al., 2025). We use Qwen3-30B-A3B (Yang et al., 2025) to translate the training data into other languages and utilize the DeepSeek-V3 (Liu et al., 2024)'s verification to further guarantee the quality of the translation. The base model trained on MGSM8K (8K samples per language) is tested on MGSM and the base model trained on LUFFY (45K samples per

Table 1: Performance of base, SFT, and RL models on MGSM. "Base" denotes Qwen2.5-3B-Base. "SFT (zh)" and "RL (zh)" indicate tuning on Chinese data. We report accuracy on 10 linguistic settings; $\Delta$ (RL–SFT) denotes the performance gap. Each value is averaged over six runs. "Avg" and "Gen" refer to the mean accuracy and generalization score, respectively.

| Models | En | Zh | De | Es | Fr | Ja | Ru | Th | Sw | Bn | Avg | Gen |
|---|---|---|---|---|---|---|---|---|---|---|---|---|
| Base | 63.4 | 48.3 | 33.5 | 57.7 | 38.9 | 19.5 | 30.3 | 17.6 | 7.3 | 1.2 | 31.8 | 0.0 |
| SFT (En) | 64.7 | 54.5 | 50.7 | 56.4 | 56.2 | 36.9 | 55.5 | 44.1 | 6.9 | 26.2 | 45.2 | 18.1 |
| RL (En) | 85.8 | 72.1 | 70.8 | 77.3 | 76.6 | 61.2 | 64.9 | 61.0 | 9.5 | 47.5 | 62.7 | 49.1 |
| $\Delta$ (RL-SFT) | +21.1 | +17.6 | +20.1 | +20.9 | +20.4 | +24.3 | +9.4 | +16.9 | +2.6 | +21.3 | +17.5 | +30.9 |
| SFT (Zh) | 65.7 | 58.7 | 48.4 | 55.7 | 56.1 | 43.5 | 56.6 | 45.8 | 7.5 | 30.5 | 46.9 | 20.4 |
| RL (Zh) | 86.1 | 76.3 | 74.2 | 81.1 | 76.1 | 64.5 | 78.1 | 64.9 | 10.3 | 48.3 | 66.0 | 52.6 |
| $\Delta$ (RL-SFT) | +20.4 | +17.6 | +25.8 | +25.4 | +20.0 | +21.0 | +21.5 | +19.1 | +2.8 | +17.8 | +19.1 | +32.3 |
| SFT (De) | 63.9 | 54.2 | 57.5 | 55.7 | 52.8 | 39.3 | 55.1 | 47.6 | 8.4 | 28.8 | 46.3 | 19.3 |
| RL (De) | 91.0 | 77.6 | 80.5 | 82.7 | 80.0 | 67.8 | 81.3 | 75.3 | 15.9 | 63.3 | 71.5 | 60.4 |
| $\Delta$ (RL-SFT) | +27.1 | +23.4 | +23.0 | +27.0 | +27.2 | +28.5 | +26.2 | +27.7 | +7.5 | +34.5 | +25.2 | +41.2 |

language) is tested on other datasets. To fully assess the reasoning ability, we evaluate the model on multilingual reasoning benchmarks from four kinds of reasoning: MGSM (Shi et al.), MMath500, and MAIME2024 (Son et al., 2025) for mathematical reasoning, MMLU-ProX-Lite (Xuan et al., 2025) for commonsense reasoning, and MGPQA-D (Qi et al., 2025) for scientific reasoning, Multilingual LogiQA (Wang et al., 2024a) which emphasizes logical reasoning. Furthermore, we use M-ifEval (Dussolle et al., 2025) to test the multilingual instruction-following capabilities.

**Learning Algorithms and Evaluation Metrics.** We compare the performance of various tuning algorithms, including SFT and RL. Specifically, we use GRPO to explore the performance of RL. The final answer is explicitly distinguished and encapsulated with in a \boxed{}. To evaluate model performance, we calculate the accuracy of each reasoning dataset. We test 6 times for MMath500 and MGSM, and 16 times for MAIME2024. This paper evaluates the reasoning capabilities of LLMs across ten languages: Bengali (Bn), Thai (Th), Swahili (Sw), Japanese (Ja), Chinese (Zh), German (De), French (Fr), Russian (Ru), Spanish (Es), and English (En). To measure the relative improvement over the base model's potential, we introduce a generalization score (Gen). This score is calculated by averaging the normalized gains across all test languages, which represents the model's ability to capitalize on the potential for improvement in each language. For a given tuned model $M_{\text{tuned}}$, the generalization score is defined as:

$$Gen(M_{\text{tuned}}) = \frac{1}{|L|} \sum_{l \in L} \frac{\text{Acc}(M_{\text{tuned}}, l) - \text{Acc}(M_{\text{base}}, l)}{1 - \text{Acc}(M_{\text{base}}, l)}$$

where $L$ is the set of evaluation languages, $\text{Acc}(M, l)$ is the accuracy of model $M$ on language $l$, and $M_{\text{base}}$ is the base model before tuning.

**Implementation Details.** All experiments utilize full-parameter tuning during both the SFT and RL phases to enable a thorough evaluation of model capabilities. The SFT experiments are carried out within the LlamaFactory (Zheng et al., 2024) framework, employing a learning rate of $2 \times 10^{-5}$, a cosine learning rate scheduler, and a batch size of 32. For RL, the verl (Sheng et al., 2024) platform is used for implementation. To guarantee a fair comparison among different RL approaches, a uniform set of parameters is adopted: the learning rate is set to $1 \times 10^{-6}$, the rollout batch size to 512, and the sampling temperature to 1.0, along with a KL-divergence coefficient of 0.001. Both SFT and RL experiments are conducted for 3 full epochs and then stop. Furthermore, we employ zero-shot setting to assess models' performance across various test datasets. To verify the robustness of our findings, we also provide 4-shots results in Table 16, which show consistent trends.

## 3.2 FINDING 1: RL EXHIBITS SUPERIOR CROSS-LINGUAL GENERALIZATION THAN SFT

**Significant performance improvement.** As shown in Table 1, RL consistently outperforms SFT across ten languages. The improvements range from +9.4 points (evaluating Russian when trained in English) to +34.5 points (evaluating Bengali when trained in German), with an overall average improvement of +17.5 to +25.2 points depending on the training language. This establishes a strong baseline for RL's superiority. The complete results are provided in Appendix A.4.1.

Table 2: Performance of base, SFT, and RL models on MMath500. We report the accuracy score on 6 linguistic settings.

| Models | Zh | Fr | En | De | Ja | Es | Avg | Gen |
|---|---|---|---|---|---|---|---|---|
| Base | 38.9 | 27.2 | 49.1 | 16.3 | 17.4 | 36.6 | 30.9 | 0.0 |
| SFT (En) | 33.6 | 56.3 | 59.7 | 56.0 | 18.1 | 57.2 | 46.8 | 22.1 |
| RL (En) | 53.7 | 55.8 | 62.7 | 50.9 | 54.2 | 56.6 | 55.7 | 34.6 |
| Δ (RL-SFT) | +20.1 | -0.4 | +3.1 | -5.1 | +36.1 | -0.6 | +8.9 | +12.5 |
| SFT (Zh) | 41.9 | 38.4 | 48.4 | 35.0 | 37.2 | 38.1 | 39.8 | 11.3 |
| RL (Zh) | 61.3 | 61.2 | 63.3 | 61.2 | 58.5 | 62.1 | 61.3 | 42.5 |
| Δ (RL-SFT) | +19.5 | +22.8 | +14.9 | +26.2 | +21.3 | +24.0 | +21.5 | +31.2 |
| SFT (De) | 31.7 | 30.7 | 38.0 | 33.5 | 19.4 | 30.3 | 30.6 | -2.6 |
| RL (De) | 61.4 | 61.5 | 62.8 | 60.7 | 60.1 | 62.1 | 61.4 | 42.6 |
| Δ (RL-SFT) | +29.7 | +30.8 | +24.8 | +27.2 | +40.7 | +31.8 | +30.8 | +45.3 |

**Robustness in cross-lingual transfer.** Notably, RL's advantage is most prominent in cross-lingual transfer scenarios, suggesting it learns more robust reasoning strategies rather than optimizing for the training language. For instance, when trained on Chinese, RL not only excels on Chinese evaluation (+17.6 points over SFT) but also generalizes significantly better to typologically distant languages like German (+25.8 points) and Spanish (+25.4 points). The consistency of these improvements across diverse language pairs (e.g., English-Bengali: +21.3 points) indicates that RL fosters the development of multilingual reasoning capabilities.

**Coherent validation.** The validity of this conclusion is further strengthened by consistent results on the MMath500 dataset in Table 2. For example, when trained on Chinese data, the RL model achieves an average accuracy of 61.3%, substantially surpassing SFT's 39.8% (a +21.5 point improvement). This cross-dataset corroboration confirms that the enhanced generalization ability of RL is not an artifact of a single benchmark.

**Effective generalization across languages in other reasoning tasks.** The superiority of RL extends beyond multilingual mathematical reasoning to challenging out-of-distribution tasks. As shown in Table 3, on benchmarks like MMLU-ProX-Lite and MGPQA-D, RL consistently maintains positive generalization scores, while SFT models often exhibit negative transfer. For instance, on MMLU-ProX-Lite, an RL model trained on German data achieves a generalization score (Gen) of 30.8, starkly contrasting with SFT's 8.0. This demonstrates that the robust reasoning representations acquired via RL are highly transferable across both linguistic and task boundaries. Notably, this advantage holds even under a double-cross setting: when trained on mathematical data in German (De) and evaluated on a commonsense reasoning task in Chinese (Zh), RL achieves a +20.0 point improvement over SFT.

**Comparison with Cold-Start.** To further validate the effectiveness of RL compared to the cold-start strategy, we conduct additional experiments using "SFT + RL" and "SFT (100 steps) + RL" settings. The detailed results are presented in Table 15. Surprisingly, we observe that directly applying RL generally yields performance superior to or comparable with the cold-start baselines. For instance, on MGSM, the average score for SFT+RL (De) is 52.6%, whereas RL (De) achieves 71.5%. This suggests that SFT might cause the model to converge prematurely to specific language patterns or local optima, thereby limiting RL's capacity to explore better strategies for multilingual reasoning.

In summary, results from cross-lingual, cross-dataset, and cross-task evaluations robustly support that RL enables models with superior generalization in multilingual reasoning compared to SFT.

### 3.3 FINDING2: RL USING NON-ENGLISH TRAINING DATA YIELDS SUPERIOR PERFORMANCE TO ENGLISH TRAINING DATA, WHILE SFT DOES NOT

**Superiority performance gains in non-English RL.** Analyzing the average performance, RL training on non-English data systematically surpasses the English baseline. Specifically, RL trained on German achieves the highest average performance at 71.5%, followed by French (70.7%) and Japanese (70.9%), shown in Table 7 in Appendix A.4.1, all substantially exceeding English-based RL training (62.7%), with the German advantage being a significant +8.8 points.

Table 3: Performance comparison on MMLU-ProX-Lite and MGPQA-D. "Avg" denotes the average score across languages (En/Zh/De), and "Gen" represents the generalization score.

| Model | MMLU-ProX-Lite | | | | | MGPQA-D | | | | |
|---|---|---|---|---|---|---|---|---|---|---|
| | En | Zh | De | Avg | Gen | En | Zh | De | Avg | Gen |
| Base | 9.2 | 2.4 | 3.6 | 5.0 | 0.0 | 21.1 | 20.0 | 20.7 | 20.6 | 0.0 |
| SFT(En) | 28.9 | 13.0 | 24.1 | 22.0 | 17.9 | 12.5 | 5.1 | 11.7 | 9.8 | -13.7 |
| RL(En) | 40.6 | 31.6 | 25.6 | 32.6 | 29.1 | 30.0 | 23.5 | 25.2 | 26.2 | 7.0 |
| Δ (RL-SFT) | +11.6 | +18.6 | +1.6 | +10.6 | +11.2 | +17.4 | **+18.4** | +13.5 | +16.4 | +20.7 |
| SFT(Zh) | 24.4 | 11.6 | 21.3 | 20.3 | 7.4 | 20.7 | 18.1 | 14.2 | 17.7 | -3.7 |
| RL(Zh) | 40.8 | 35.0 | 34.4 | 36.7 | 33.4 | 25.0 | 27.3 | 28.3 | 26.9 | 7.8 |
| Δ (RL-SFT) | +16.3 | **+23.4** | +13.1 | +16.4 | +19.8 | +4.3 | +9.2 | +14.1 | +9.2 | +11.5 |
| SFT(De) | 15.8 | 7.3 | 15.0 | 12.7 | 8.0 | 7.2 | 12.8 | 8.8 | 9.6 | -13.9 |
| RL(De) | 39.9 | 27.2 | 35.5 | 34.2 | 30.8 | 26.2 | 27.2 | 25.3 | 26.2 | 7.1 |
| Δ (RL-SFT) | **+24.1** | +20.0 | **+20.5** | **+21.5** | **+22.8** | **+19.0** | +14.4 | **+16.6** | **+16.7** | **+21.0** |

Further more, the superiority is also pronounced in cross-lingual scenarios. For example, RL trained on German not only excels on German evaluation (80.5%) but also shows remarkable transfer to distant languages like Bengali (63.3% vs 47.5% for English, +15.8 pts) and Thai (75.3% vs 61.0% for English, +14.3 pts), indicating learning of transferable representations.

**Similar phenomenon across diverse tasks.** The pattern is consistently validated on diverse benchmarks. On MMLU-Pro-X Lite, from Table 3, RL trained on Chinese achieves 36.7%, outperforming RL trained on English (32.6%). On MGSM, RL trained on German attains a generalization score of +41.2, significantly higher than RL trained on English (+30.9), confirming the robust and generalizable benefits of non-English RL training. Moreover, as shown in Table 17 and Table 18, performance on M-ifEval and Multilingual LogiQA, which assess instruction following and logical reasoning, consistently demonstrates that non-English RL yields superior cross-lingual generalization.

**The Different Phenomenon observed in SFT.** This phenomenon appears exclusively with RL. In contrast, SFT results exhibit minimal variation across training languages, with Avg scores ranging from 46.3% on German to 47.6% on Japanese (see Table 7 in Appendix A.4.1). The performance differences remain within statistical noise, ruling out data quality as the sole explanation and highlighting the critical role of the RL objective.

**Comparison with Mixed-Language Training.** To further investigate the performance of training on mixed languages, we use a mixture of English, Chinese, and German training data (Mix), ensuring the total data volume remains consistent. Table 14 shows that while RL (Mix) achieves competitive results (Average 68.1%), RL (De) still maintains the highest performance (Average 71.5%). This phenomenon indicates that some specific non-English languages can stimulate the model's generalization potential in RL training more effectively than even mixing multiple languages.

### 3.4 SAME PHENOMENON ON ANOTHER BASE MODEL

In Table 4, we report the performance of SmolLM3-3B-Base under the same configuration of Qwen2.5-3B-Base. We find that our observations are consistently.

**Finding 1: RL exhibits superior cross-lingual generalization than SFT.** Across all training languages, RL consistently and significantly outperforms SFT. The improvements are substantial, ranging from +11.7 points (evaluating Swedish when trained on German) to +34.7 points (evaluating Chinese when trained on German).

**Finding 2: RL using non-English training data yields superior performance to English training data, while SFT does not.** Similar to the trend observed on Qwen2.5-3B-Base, RL trained on non-English data surpasses RL trained on English. RL (De) reaches the highest average accuracy at 69.9 and the strongest generalization score at 64.9. In contrast, SFT models remain far behind.

To ensure the robustness of our findings across model scales, we further verify our conclusions on Qwen2.5-7B-Base. As shown in Table 13, 7B model exhibits a trend highly consistent with the 3B model: (1) RL achieves significantly higher performance gains compared to SFT. (2) RL training on non-English data (e.g., German, Chinese) continues to demonstrate stronger cross-lingual general-

Table 4: Performance of base model, SFT, and RL tuning models on MGSM. Base denotes the original SmolLM3-3B-Base model. We report the accuracy score on 10 linguistic settings.

| Models | En | Zh | De | Es | Fr | Ja | Ru | Th | Sw | Bn | Avg | Gen |
|---|---|---|---|---|---|---|---|---|---|---|---|---|
| Base | 29.3 | 17.1 | 24.3 | 27.3 | 26.9 | 18.1 | 24.1 | 13.0 | 4.0 | 5.7 | 19.0 | 0.0 |
| SFT (En) | 60.3 | 40.7 | 49.3 | 51.8 | 48.0 | 32.8 | 47.7 | 37.7 | 7.3 | 10.0 | 38.6 | 25.3 |
| RL (En) | 87.3 | 70.8 | 78.3 | 81.0 | 81.3 | 62.2 | 79.7 | 67.3 | 16.5 | 22.2 | 64.6 | 58.6 |
| Δ (RL-SFT) | +27.1 | +30.1 | **+28.9** | +29.2 | **+33.3** | +29.4 | +32.0 | +29.5 | +9.1 | +12.2 | +26.1 | +33.3 |
| SFT (Zh) | 58.9 | 51.5 | 50.5 | 54.9 | 54.9 | 39.3 | 48.7 | 43.7 | 9.2 | 11.7 | 42.3 | 30.0 |
| RL (Zh) | 88.6 | 77.1 | 79.3 | 82.5 | 80.9 | 71.1 | 82.4 | 76.2 | 20.1 | 28.5 | 68.7 | 63.4 |
| Δ (RL-SFT) | **+29.7** | +25.6 | +28.7 | +27.6 | +26.0 | **+31.8** | +33.7 | +32.5 | +10.9 | +16.9 | +26.3 | +33.4 |
| SFT (De) | 60.1 | 43.5 | 53.9 | 56.1 | 52.1 | 38.3 | 51.8 | 43.3 | 9.0 | 9.7 | 41.8 | 29.4 |
| RL (De) | 85.1 | 78.2 | 81.7 | 85.6 | 84.1 | 69.1 | 85.8 | 77.2 | 20.7 | 31.3 | 69.9 | 64.9 |
| Δ (RL-SFT) | +24.9 | **+34.7** | +27.7 | **+29.5** | +31.9 | +30.8 | **+34.0** | **+33.9** | **+11.7** | **+21.7** | **+28.1** | **+35.5** |

ization capabilities than RL on English data. This provides compelling evidence that the superiority of non-English RL is not specific to small models but holds at larger parameter scales.

# 4 MECHANICS ANALYSIS OF RL'S CROSS-LINGUAL GENERALIZATION

To investigate why RL exhibits stronger generalization than SFT, and why RL training on non-English data outperforms that on English, we present an abbreviated set of responses generated by RL-trained models on the test set in Table 5. The complete responses are provided in Appendix A.5. Our analysis reveals that when models are trained using German instructions during RL training, the resulting models do not strictly adhere to German when generating thinking and responses. Instead, they employ non-German or mixed languages for reasoning processes. This observation attracts our attention and leads us to propose a hypothesis: could this inconsistent language usage in reasoning contribute to the enhanced generalization observed in RL training?

## 4.1 EXPLORATION OF LANGUAGE CONSISTENCY IN RL

To empirically validate this hypothesis, we conduct comparative experiments using two distinct approaches: (1) employing prompts that strictly constrain language usage, and (2) incorporating the language consistency reward that encourages the model to adhere to the language of the question into the RL training process. The details are as follows:

$$r_{\text{overall}} = 0.5 r_{\text{acc}} + 0.5 r_{\text{consistency}} \tag{1}$$

The language consistency reward $r_{consistency}$ is designed to explicitly encourage the model's output language to match the input instruction's language. We implement this using langid (Lui & Baldwin) as a language identifier to detect the primary language of the generated response. A positive reward is given for a match, and a penalty is applied otherwise.

We investigate the impact of language consistency by forcing the model to use a specific language during inference and observing performance changes. The results are presented in Figure 2 and Table 6, with a specific case study provided in Table 5.

We observe two key aspects: (a) Language inconsistency serves as a potential source of cross-lingual generalization capability, and (b) Building upon this mechanism, RL (De) exhibits greater language inconsistency than RL (En), resulting in superior cross-lingual performance. These observations suggest that the degree of flexibility in deviating from training language constraints may determine the extent of cross-lingual generalization achieved by RL-trained models.

**Language inconsistency enhances cross-lingual generalization.** Enforcing language consistency significantly degrades performance. As demonstrated in Figure 2, both RL (Zh) and RL (De) models achieve strong baseline performance (61.3% and 61.4%, respectively). However, when constrained to use their training languages through prompting, performance drops substantially—RL (Zh) falls to 53.7% and RL (De) to 60.5%. The degradation becomes even more pronounced when language consistency rewards are applied, with RL (De) plummeting to 52.0%. This pattern strongly indicates that enforced language consistency impairs cross-lingual reasoning capabilities.

Table 5: Case study of on MMath500. We present a case of German question.

> **Question:** Ein Matheclub veranstaltet einen Kuchenverkauf als Spendenaktion, um Geld für eine bevorstehende Reise zu sammeln. Sie verkaufen 54 Kekse zu drei für $1$, und $20$ Cupcakes zu $2$ pro Stück, und $35$ Brownies zu $1$ pro Stück. Wenn es den Matheclub $15$ gekostet hat, diese Artikel zu backen, wie hoch war ihr Gewinn?
> **Answer**: 78
>
> **Base:**
> Da Kekse zu drei für $1 verkaufen werden, müssen wir zahlenweise Kekse zählen. Es gibt 54 Kekse zu backen, und jeder Kuchen besteht aus drei Kekse. Also haben sie $54/3 = 18$ Kuchen zu backen. ... Da sie für die Backzeit 15 Dollar ausgeben, haben sie für den Verkauf $146 - 15 = \$131$ eingenommen. Daher haben sie einen Gewinn von $\boxed{131}$.
>
> **Reinforcement Learning (with German Data):**
> 1. Calculate the revenue from Kekse: - They sold 54 Kekse at a rate of 3 Kekse for $1. - The number of sets of 3 Kekse in 54 is $\frac{54}{3} = 18$. - Revenue from Kekse $= 18 \times 1 = 18$ dollars.
> ...
> 5. Calculate the profit: - Cost to make the items = $15. - Profit = Total revenue - Cost to make the items - Profit $= 93 - 15 = 78$ dollars.
> Therefore, the profit is $\boxed{78}$.
>
> **Reinforcement Learning (with German Data and with Consistency Reward):**
> Zuerst berechnen wir den Einnahmen von den Keksen. Da sie drei Kekse für $1 verkaufen, erhalten sie: $54 \div 3 = 18$ Doppelsätze, ... Da es ihnen $15 gekostet hat, um die Artikel zu backen, erhalten sie: $91 - 15 = 76$ Dollar Die Gewinnsumme beträgt $\boxed{76}$.

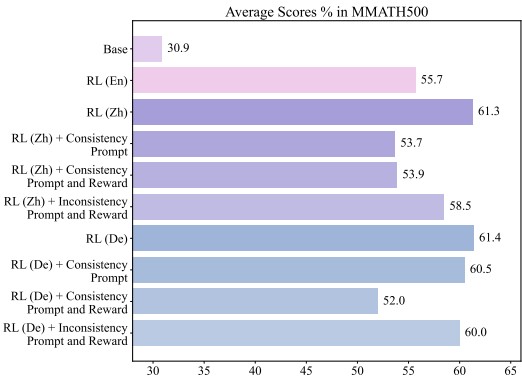

Figure 2: Scores on MMath500. The chart compares the average accuracy of different models. "RL (Zh)" indicates training on Chinese data.

Table 6: Language consistency of models on MMath500. We test 6 times and report the average percentage of language consistency.

| Models | En (%) | Zh (%) | De (%) |
|---|---|---|---|
| Base | 99.4 | 91.4 | 94.5 |
| SFT (En) | 98.6 | 99.3 | 83.7 |
| RL (En) | 99.9 | 89.0 | 96.2 |
| SFT (Zh) | 99.7 | 98.9 | 81.3 |
| RL (Zh) | 99.8 | 0.0 | 0.0 |
| + Consistency Prompt | 99.3 | 99.6 | 97.3 |
| + Consistency Prompt and Reward | 99.9 | 99.8 | 98.9 |
| + Inconsistency Prompt and Reward | 99.7 | 0.0 | 8.1 |
| SFT (De) | 94.2 | 85.5 | 99.1 |
| RL (De) | 99.7 | 4.8 | 0.0 |
| + Consistency Prompt | 99.8 | 52.4 | 0.0 |
| + Consistency Prompt and Reward | 99.8 | 99.8 | 99.9 |
| + Inconsistency Prompt and Reward | 99.6 | 0.0 | 0.0 |

Table 6 reveals that unconstrained RL models show low consistency in their training languages—both RL (Zh) and RL (De) achieve 0.0% consistency, indicating they spontaneously adopt other languages during reasoning. Conversely, constrained models exhibit high consistency rates (up to 99.9% for RL (De) with consistency rewards), but at the cost of reduced performance. This negative correlation between language consistency and performance suggests that linguistic flexibility enables models to leverage more powerful, multilingual reasoning modules.

**Case analysis of the language inconsistency.** As shown in Table 5, When solving German question, the unconstrained model (RL (De)) employs mixed English and German reasoning and reaches correct solutions, while the consistency-constrained model, despite only reasoning in German, produces flawed logical steps and wrong answers. This demonstrates that constraining models to specific languages may inhibit access to more robust reasoning patterns established during pre-training.

**Language inconsistency in non-English RL.** The performance comparison in Figure 2 shows that RL (De) achieves 61.4% average accuracy compared to RL (En)'s 55.7%. More importantly, RL (De) maintains strong performance across different target languages, while RL (En) shows more pronounced degradation in non-English tasks. This suggests that German-based training may provide advantages for cross-lingual generalization.

Different source languages yield distinct generalization patterns. The superior performance of RL (De) may stem from German's grammatical complexity and its linguistic distance from other languages, potentially encouraging the development of more multilingual reasoning strategies. In con-

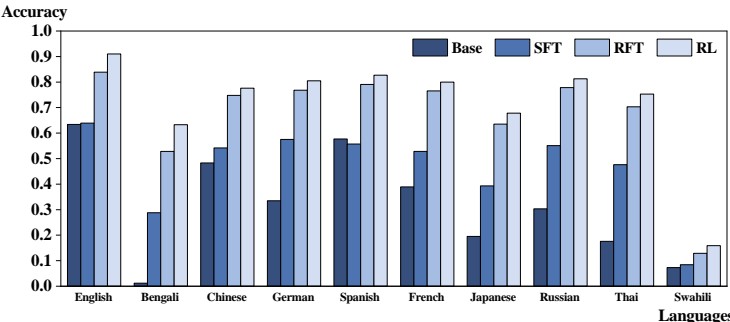

Figure 3: Model performance comparisons among the Base, SFT, RFT, and RL models on MGSM. We use German data in LUFFY in SFT, RL, RFT for training.

trast, English-based training might lead to more language-specific reasoning patterns that transfer less effectively across languages.

**Language consistency in SFT.** Unlike RL models, SFT models exhibit high language consistency (e.g., SFT (Zh) maintains 99.7% consistency) due to their imitation-based training paradigm. While this consistency appears desirable, it actually constrains generalization by trapping models within language-specific thought patterns established during training, leading to impaired cross-lingual performance when facing problems in other languages.

Furthermore, results in Figure 2 show that while encouraging inconsistency (RL + Inconsistency Prompt and Reward) yields better performance than enforcing consistency (RL + Consistency Prompt and Reward), allowing the RL model to autonomously select the language (RL) still achieves the best results. This suggests that while language inconsistency is a key factor in the superiority of RL, freely exploring reasoning paths without forced constraints is also crucial.

## 4.2 Exploration of sampling in RL

To further investigate the source of RL's advantage over SFT, we analyze the role of sampling in performance enhancement. We introduce Rejection Sampling Fine-Tuning (RFT) (Touvron et al., 2023) as an intermediate baseline between SFT and full RL. The RFT we use involves sampling multiple times from the model after RL training. It then fine-tunes the model using only the samples that yield the correct answer. This represents a more on-policy exploration mechanism than SFT.

As shown in Figure 3, across all languages, accuracy increases progressively from the Base model to SFT, RFT, and finally to the RL-tuned model. Specifically, SFT achieves 46.3% average accuracy, RFT improves to 66.8%, and RL reaches 71.5%. This trend underscores the importance of the model's exploration of solution paths in enhancing its reasoning abilities.

**Better Performance with Data Aligned to the Model's Distribution.** Although SFT follows a completely correct off-policy solution path, RFT data, more aligned with the model's distribution, enables the model to explore reasoning chains better suited to its own configuration through sampling. This alignment helps the model capture reasoning patterns and optimization trajectories more effectively, allowing it to generalize beyond memorized solutions.

**The Importance of Online Optimization in RL.** Compared to RFT, RL (GRPO in our experiments) continuously performs more on-policy sampling with both positive and negative examples during training. This not only further aligns the data with the model but also goes beyond mere imitation learning. As shown in Figure 5, RL consistently outperforms the other methods across all languages, demonstrating that the online policy optimization process in RL is more effective at enhancing the model's generalizable reasoning than RFT.

**Uncertainty Promotes Cross-Lingual Exploration.** To further investigate why RL training on German data yields superior transferability, we analyze the sampling diversity. We employ the base model to generate six responses for each question across different languages via sampling. We then calculate the Perplexity (PPL) and Self-Similarity (measured by BLEU scores among sampled responses for each question) of the base model's outputs across different languages. As shown in

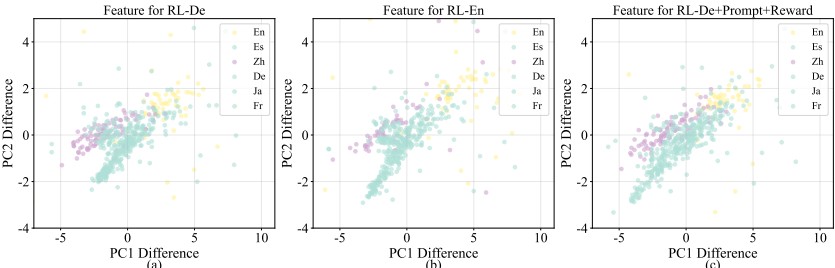

Figure 4: Feature of LLM's hidden state of last layer, in training (dataset LUFFY) configuration of (a)RL-De, (b)RL-De+Prompt, and (c))RL-De+Prompt+Reward. "+Prompt" adds language control prompts, and "+Reward" adds a language consistency reward.

Table 11, German questions exhibit higher PPL (1.414) and and the lowest Self-Similarity (0.425). This indicates that the model faces higher uncertainty when processing German questions. In the exploration phase of RL, this uncertainty potentially prompts the model to step out of single-language constraints and explore reasoning paths in mixed languages or its dominant language (English). This process inadvertently activates stronger cross-lingual generalization capabilities. In contrast, the low perplexity in English questions limits this diversity during sampling. Furthermore, SFT tends to closely fit the language distribution, also limiting such exploration.

### 4.3 EXPLORATION OF MODEL SEMANTIC FEATURE SHIFT

To investigate why RL training with different languages yields varying generalization capabilities, we analyze the semantic feature shifts in learned representations.

**Methodology.** We extract final layer hidden states from base and RL-trained models when processing MMath500 test data across six languages. These representations are projected to 2D space using PCA, and we compute difference vectors: $\mathbf{h}_{diff} = \mathbf{h}_{RL} - \mathbf{h}_{Base}$.

**Results.** Figure 4 reveals distinct shift patterns across training configurations. RL-De exhibits the most concentrated distribution around the origin, indicating minimal deviation from base representations, while RL-En displays more scattered distributions. This ordering directly correlates with cross-lingual performance in Table 2. Similarly, language consistency interventions in RL-De progressively increase representational scatter: baseline RL-De maintains compact distributions, while RL-De+Prompt+Reward shows greater dispersion, mirroring the performance degradation pattern.

**Interpretation.** These findings suggest that pre-training establishes multilingual reasoning structures crucial for cross-lingual transfer (Hua et al., 2024; Merchant et al., 2020). Models preserving these structures through minimal representational drift maintain stronger generalization capabilities. Conversely, larger shifts disrupt universal reasoning mechanisms (Luo et al., 2025), explaining why RL's linguistic inconsistency paradoxically enhances cross-lingual performance by preserving pre-trained reasoning structures (Lai et al., 2025).

## 5 CONCLUSION

We systematically investigated the differences between Reinforcement Learning and Supervised Fine-Tuning for enhancing cross-lingual reasoning and the generalization across languages. Multiple experiments demonstrate that RL not only achieves substantially higher accuracy than SFT but also exhibits superior cross-lingual generalization. Contrary to conventional cognition, we find that RL training on non-English data yields superior performance, challenging English-centric training. Our preliminary mechanistic analysis investigates the potential reasons for the superior cross-lingual generalization of RL from three perspectives: the linguistic inconsistency during the reasoning process, the unique explore-and-optimize sampling strategy, and the semantic shift after training. The understanding of these potential factors not only provides crucial insights into understanding RL's advantages in multilingual reasoning but also establishes a foundation for effectively enhancing cross-lingual reasoning in the future.

ACKNOWLEDGMENTS

This work has been financially supported by the National Key R&D program of China No. 2022YFE0204900 and the National Natural Science Foundation of China (NSFC) Key Project under Grant Number 62336006.

ETHICS STATEMENT

This study acknowledges several ethical implications of its investigation into cross-lingual reasoning in LLMs. Data and fairness concerns arise from potential biases in multilingual benchmarks, which may introduce performance disparities across languages. Our evaluations incorporate diverse linguistic settings and different multilingual reasoning tasks to mitigate such biases, though future work must further scrutinize cultural and linguistic influences on model behavior.

Beyond technical limitations, societal impact requires careful consideration. While improved multilingual reasoning could enhance accessibility for non-English speakers, reducing barriers in education and professional settings, it also risks misuse—such as automated disinformation generation or harmful content propagation across languages. We advocate for responsible deployment, emphasizing robust safeguards, human oversight, and ongoing risk assessments to balance innovation with ethical constraints.

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

# A APPENDIX

## A.1 USE OF LLM

During the preparation of this work, large language models (e.g., ChatGPT) were used for English writing refinement and minor assistance in code debugging. All ideas, experiments, and analyses are solely by the authors.

## A.2 EXPERIMENTAL SETTINGS

### A.2.1 RFT SETTINGS

In the RFT experiments, we sample from the RL-trained model on the training set with a temperature of 1.0 and top-p of 0.95. For each prompt, we sample 10 times and filter for responses with correct answers for fine-tuning. We strictly controll the RFT training data volume to be consistent with SFT.

### A.2.2 PROMPTS FOR GENERATING TRANSLATED GSM8K TRAINING DATASETS

Here is the prompts for generating translated GSM8K training datasets.

```
TRANSLATION_PROMPT_TEMPLATE = """You are a professional math
translation assistant. Please translate the following English
math problem into {target_language}, maintaining the mathematical
expressions and formatting.

Requirements:
1. Maintain the format of the mathematical calculation process
(e.g., <<48/2=24>>)
2. Maintain the format of the final answer (e.g., #### 72)
3. The translation should be accurate and natural.
4. Keep the numbers and mathematical symbols unchanged.

Original question:

Original answer:

Please translate the question and answer separately,
using the following format:
{{
"translated_question": "Translated question",
"translated_answer": "Translated answer"
}}

Please return the result in JSON format, using the
{target_language} language, and do not add any additional text.
"""
```

### A.2.3 PROMPTS FOR GENERATING TRANSLATED LUFFY TRAINING DATASETS

Here is the prompts for generating translated LUFFY training datasets.

```
TRANSLATION_PROMPT_TEMPLATE = """You are a professional math
translation assistant. Please translate the following content
into {target_language}, preserving mathematical expressions,
LaTeX formulas, and special formatting.

Requirements:
1. Keep all mathematical formulas and LaTeX expressions
intact (e.g., $24 \\mathrm{{~km}}$, \\boxed{{}}, etc.)
2. Keep the <think> and </think> tags intact
3. The translation should be accurate and natural.
4. Keep numbers and mathematical symbols intact.

Original content:
{content}

Please return the translated {target_language} content
directly in the format
{{
"translated_content": "Translated content"
}}
Do not add any additional explanatory text.

"""
```

### A.3 TRANSLATION DETAILS

To ensure high translation quality, we implement an automated verification mechanism during the translation process. Instances that fail this verification are re-generated until they met the quality standards.

To further verify translation quality, we sample translation examples from each language and use DeepSeek-V3.2 (Liu et al., 2024), Deepseek-R1 (Guo et al., 2025), and GPT-4o (Achiam et al., 2023) as judges to conduct a head-to-head quality comparison between our translations and the validated MGSM8K-Instruct dataset (Chen et al., 2024). We utilize the first 20 strictly aligned examples from MGSM8K-Instruct (20 per language) to conduct a direct comparison with our corresponding translated data. The results are shown in Table 12. Our translation data has a win rate comparable to MGSM8K-Instruct (Average Wins: Ours 47.2% vs MGSM8K 45.0%), demonstrating the high quality of our training data, which is comparable to the high-quality MGSM8K-Instruct dataset.

## A.4 RESULTS DETAILS

### A.4.1 LANGUAGE USAGE STATISTICS

To further understand the model's behavior, we also analyze the language usage statistics detailed in Table 19. We conduct a detailed language analysis on MMath500 by randomly sampling 100 questions for each language (En, Zh, De, Es, Fr, Ja). For each question, we sample 6 responses and calculate the rank score of language usage to analyze the distribution. We report the scores for the primary languages (En, Zh, De, Es, Fr, Ja) rank scores across different models. Specifically, for each response, we use DeepSeek-v3.2 (Liu et al., 2024) to annotate the primary languages used, identifying up to 3 languages per response. The scoring rules are as follows: (1) Rank 1 language receives a score of 1. (2) Rank 2 language receives a score of 1/2. (3) Rank 3 language receives a score of 1/3. (4) If only one primary language is identified, it receives a score of 1. Results show that the Base Model's language usage is relatively balanced. After SFT and RL training, the usage of English significantly increases. RL-trained models (especially RL-Zh and RL-De) show a significant increase in the usage of English (or English-mixed language) when answering non-English questions. The RL training process enables the model to adaptively learn and select suitable languages for complex reasoning, rather than passively adhering to the input language. With the Inconsistency Analysis, this diversity and adaptive selection capability confer better performance to RL, facilitating more effective Cross-Lingual Transfer.

Table 7: Performance of base model, SFT, and RL tuning models on MGSM. Base denotes the original Qwen2.5-3B-Base model. SFT (zh) and RL (zh) mean we tune the base model in Chinese data through SFT and RL, respectively. We report the accuracy score on 10 linguistic settings. $\Delta$ (RL-SFT) represents the performance difference between RL and the corresponding SFT score. Each score represents the average accuracy over six measurements. Avg represents the average of the scores of 10 language settings and Gen represents the generalization score.

| Models | En | Zh | De | Es | Fr | Ja | Ru | Th | Sw | Bn | Avg | Gen |
|---|---|---|---|---|---|---|---|---|---|---|---|---|
| Base | 63.4 | 48.3 | 33.5 | 57.7 | 38.9 | 19.5 | 30.3 | 17.6 | 7.3 | 1.2 | 31.8 | 0.0 |
| SFT (En) | 64.7 | 54.5 | 50.7 | 56.4 | 56.2 | 36.9 | 55.5 | 44.1 | 6.9 | 26.2 | 45.2 | 18.1 |
| RL (En) | 85.8 | 72.1 | 70.8 | 77.3 | 76.6 | 61.2 | 64.9 | 61.0 | 9.5 | 47.5 | 62.7 | 49.1 |
| $\Delta$ (RL-SFT) | +21.1 | +17.6 | +20.1 | +20.9 | +20.4 | **+24.3** | +9.4 | +16.9 | +2.6 | *+21.3* | +17.5 | +30.9 |
| SFT (Zh) | 65.7 | 58.7 | 48.4 | 55.7 | 56.1 | 43.5 | 56.6 | 45.8 | 7.5 | 30.5 | 46.9 | 20.4 |
| RL (Zh) | 86.1 | 76.3 | 74.2 | 81.1 | 76.1 | 64.5 | 78.1 | 64.9 | 10.3 | 48.3 | 66.0 | 52.6 |
| $\Delta$ (RL-SFT) | +20.4 | +17.6 | **+25.8** | **+25.4** | +20.0 | +21.0 | +21.5 | +19.1 | +2.8 | +17.8 | +19.1 | +32.3 |
| SFT (De) | 63.9 | 54.2 | 57.5 | 55.7 | 52.8 | 39.3 | 55.1 | 47.6 | 8.4 | 28.8 | 46.3 | 19.3 |
| RL (De) | 91.0 | 77.6 | 80.5 | 82.7 | 80.0 | 67.8 | 81.3 | 75.3 | 15.9 | 63.3 | 71.5 | 60.4 |
| $\Delta$ (RL-SFT) | *+27.1* | +23.4 | +23.0 | +27.0 | +27.2 | +28.5 | +26.2 | +27.7 | **+7.5** | **+34.5** | **+25.2** | ***+41.2*** |
| SFT (Es) | 63.9 | 54.7 | 54.3 | 62.7 | 54.0 | 41.1 | 58.1 | 46.5 | 9.5 | 31.1 | 47.6 | 21.6 |
| RL (Es) | 89.3 | 77.8 | 78.0 | 82.1 | 77.3 | 68.9 | 80.3 | 72.7 | 13.4 | 53.7 | 69.4 | 57.5 |
| $\Delta$ (RL-SFT) | +25.4 | +23.1 | +23.7 | +19.4 | +23.3 | +27.8 | +22.2 | +26.2 | +3.9 | +22.6 | +21.8 | +35.9 |
| SFT (Fr) | 64.8 | 53.7 | 51.3 | 58.8 | 57.9 | 40.9 | 57.1 | 46.5 | 8.9 | 29.9 | 47.0 | 20.6 |
| RL (Fr) | 89.3 | 78.9 | 77.5 | 82.3 | 81.1 | 70.9 | 81.1 | 73.1 | 13.3 | 59.1 | 70.7 | 59.3 |
| $\Delta$ (RL-SFT) | +24.5 | +25.2 | +26.2 | +23.5 | +23.2 | **+30.0** | +24.0 | +26.6 | +4.4 | +29.2 | +23.7 | +38.7 |
| SFT (Ja) | 64.4 | 56.5 | 50.5 | 58.2 | 53.6 | 51.3 | 54.3 | 45.6 | 8.1 | 33.7 | 47.6 | 21.1 |
| RL (Ja) | 88.1 | 79.1 | 78.8 | 81.5 | 79.3 | 72.7 | 81.7 | 72.4 | 14.1 | 61.7 | 70.9 | 59.2 |
| $\Delta$ (RL-SFT) | +23.7 | +22.6 | +28.3 | +23.3 | +25.7 | +21.4 | **+27.4** | +26.8 | +6.0 | +28.0 | +23.3 | +38.1 |
| SFT (Ru) | 64.8 | 54.9 | 53.5 | 56.7 | 55.1 | 39.5 | 57.3 | 44.9 | 10.4 | 29.8 | 46.7 | 20.0 |
| RL (Ru) | 87.5 | 76.6 | 78.5 | 79.9 | 78.8 | 69.6 | 80.3 | 73.5 | 12.5 | 57.8 | 69.5 | 57.1 |
| $\Delta$ (RL-SFT) | +22.7 | +21.7 | +25.0 | +23.2 | +23.7 | +30.1 | +23.0 | **+28.6** | +2.1 | +28.0 | +22.8 | +37.1 |

Table 8: Performance of base model, SFT, and RL tuning models on MAIME2024. Base denotes the original Qwen2.5-3B-Base model. SFT (zh) and RL (zh) mean we tune the base model in Chinese data through SFT and RL, respectively. We report the Pass@16 score on 10 linguistic settings. $\Delta$ (RL-SFT) represents the performance difference between RL and the corresponding SFT score.

| Models | Zh | Fr | En | De | Ja | Es | Ru | Th | Bn | Sw | Average |
|---|---|---|---|---|---|---|---|---|---|---|---|
| Base | 6.7 | 6.7 | 16.7 | 10.0 | 10.0 | 13.3 | 3.3 | 10.0 | 3.3 | 0.0 | 8.0 |
| SFT (Zh) | 20.0 | 6.7 | 13.3 | 6.7 | 10.0 | 13.3 | 10.0 | 20.0 | 6.7 | 3.3 | 11.0 |
| RL (Zh) | 26.7 | 30.0 | 23.3 | 26.7 | 26.7 | 23.3 | 23.3 | 26.7 | 20.0 | 10.0 | 23.7 |
| $\Delta$ (RL-SFT) | +6.7 | +23.3 | +10.0 | +20.0 | +16.7 | +10.0 | +13.3 | +6.7 | +13.3 | +6.7 | +12.7 |
| SFT (De) | 13.3 | 13.3 | 16.7 | 13.3 | 0.0 | 6.7 | 10.0 | 3.3 | 6.7 | 6.7 | 9.0 |
| RL (De) | 26.7 | 20.0 | 20.0 | 23.3 | 13.3 | 30.0 | 26.7 | 20.0 | 13.3 | 16.7 | 21.0 |
| $\Delta$ (RL-SFT) | +13.4 | +6.7 | +3.3 | +10.0 | +13.3 | +23.3 | +16.7 | +16.7 | +6.6 | +10.0 | +12.0 |

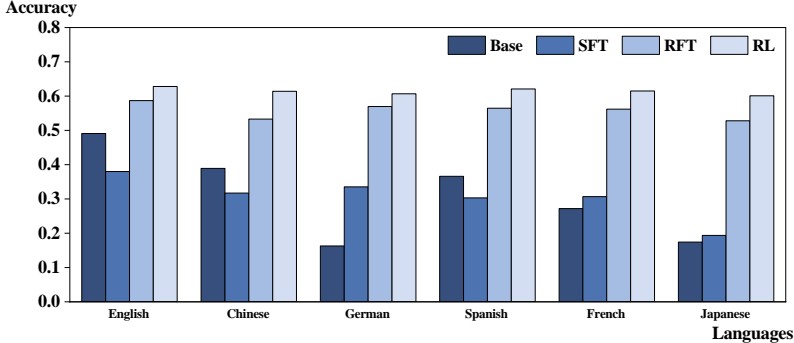

(a) A comparison of performance on MMath500.

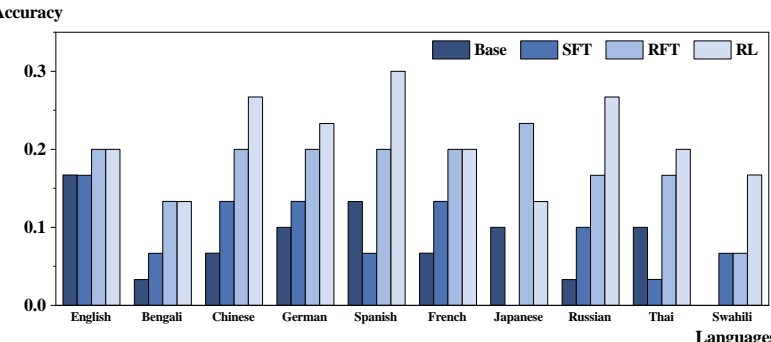

(b) A comparison of performance on MAIME2024.

Figure 5: Model performance comparisons among the Base, SFT, RFT, and RL models. We use German data in LUFFY in SFT, RL, RFT for training.

Table 9: Performance of models on MMath500. "RL (zh)" denotes the model trained with Reinforcement Learning on Chinese data. "+ Consistency Prompt" indicates the addition of language control prompts during both the training and the inference. "+ Consistency Prompt and Reward" further incorporates a language consistency reward into the training objective. "+ Inconsistency Prompt and Reward" incorporates the inconsistency prompt and inconsistency reward into the training objective. We report the accuracy score on 6 linguistic settings. We test 6 times and report the average accuracy scores and pass@k scores.

| Models | Zh | Fr | En | De | Ja | Es | Average |
|---|---|---|---|---|---|---|---|
| **Average Scores** | | | | | | | |
| Base | 38.9 | 27.2 | 49.1 | 16.3 | 17.4 | 36.6 | 30.9 |
| RL (En) | 53.7 | 55.8 | 62.7 | 50.9 | 54.2 | 56.6 | 55.7 |
| RL (Zh) | 61.3 | 61.2 | 63.3 | 61.2 | 58.5 | 62.1 | 61.3 |
| + Consistency Prompt | 53.3 | 54.9 | 59.7 | 42.2 | 55.1 | 56.8 | 53.7 |
| + Consistency Prompt and Reward | 56.2 | 54.2 | 62.9 | 45.5 | 48.2 | 56.4 | 53.9 |
| +Inconsistency Prompt and Reward | 59.9 | 56.1 | 61.6 | 57.6 | 57.7 | 58.2 | 58.5 |
| RL (De) | 61.4 | 61.5 | 62.8 | 60.7 | 60.1 | 62.1 | 61.4 |
| + Consistency Prompt | 56.0 | 61.3 | 63.8 | 60.8 | 59.0 | 61.9 | 60.5 |
| + Consistency Prompt and Reward | 51.9 | 52.1 | 62.4 | 49.3 | 41.6 | 54.6 | 52.0 |
| +Inconsistency Prompt and Reward | 59.1 | 60.1 | 62.4 | 60.1 | 57.5 | 60.9 | 60.0 |
| **Pass@6 Scores** | | | | | | | |
| Base | 67.9 | 60.9 | 75.8 | 48.1 | 44.9 | 69.3 | 61.2 |
| RL (En) | 74.7 | 77.2 | 78.8 | 73.9 | 75.4 | 77.6 | 76.3 |
| RL (Zh) | 78.4 | 76.2 | 81.4 | 78.0 | 75.2 | 77.8 | 77.8 |
| + Consistency Prompt | 73.9 | 76.8 | 77.0 | 72.7 | 74.5 | 77.2 | 75.4 |
| + Consistency Prompt and Reward | 74.1 | 73.7 | 81.4 | 72.7 | 70.9 | 76.2 | 74.8 |
| +Inconsistency Prompt and Reward | 77.8 | 76.6 | 78.8 | 76.8 | 77.2 | 76.2 | 77.2 |
| RL (De) | 78.4 | 78.8 | 79.0 | 76.4 | 77.8 | 78.8 | 78.2 |
| + Consistency Prompt | 77.8 | 79.4 | 80.4 | 77.8 | 77.8 | 79.2 | 78.7 |
| + Consistency Prompt and Reward | 74.1 | 73.9 | 79.0 | 72.9 | 65.3 | 76.2 | 73.6 |
| +Inconsistency Prompt and Reward | 78.6 | 77.6 | 80.4 | 78.4 | 76.8 | 77.8 | 78.3 |

To complement Figure 4, Table 10 reports the quantitative measurements of representational movement under different RL configurations. Specifically, "Model Center Distance" denotes the distance between each model's representation center and the base model center, while "Model Shift Distance" denotes the distance between the model's shift center and the zero point. These measurements provide quantitative evidence supporting the representational patterns illustrated in the figure.

Table 10: Numerical results corresponding to Figure 4, reporting the model center distance and shift distance under different RL configurations.

| Config | Model Center Distance | Model Shift Distance |
|---|---|---|
| RL-En | 2.255 | 41.332 |
| RL-Zh | 1.815 | 41.294 |
| RL-De | 1.753 | 41.241 |
| RL-De+Prompt | 1.891 | 41.286 |
| RL-De+Prompt+Reward | 1.908 | 41.652 |

Table 11: Sampling of the Base Model on different language questions. We calculate the average Perplexity of responses, and Self-Similarity among responses for each question of six sampling times.

|                 | En    | Zh    | De    | Fr    | Ja    | Es    |
|-----------------|-------|-------|-------|-------|-------|-------|
| Perplexity      | 1.186 | 1.248 | 1.414 | 1.332 | 1.440 | 1.267 |
| Self-Similarity | 0.621 | 0.505 | 0.425 | 0.448 | 0.433 | 0.534 |

Table 12: The translation quality comparison between ours and MGSM8K-Instruct.

| Languages | Ours Wins Ratio (%) | MGSM8K-Instruct Wins Ratio (%) | Ties Ratio (%) | Fleiss' Kappa |
|-----------|---------------------|--------------------------------|----------------|---------------|
| Zh        | 63.3                | 33.3                           | 3.3            | 0.726         |
| De        | 46.7                | 48.3                           | 5.0            | 0.817         |
| Fr        | 40.0                | 50.0                           | 10.0           | 0.885         |
| Ja        | 48.3                | 46.7                           | 5.0            | 0.756         |
| Es        | 36.7                | 46.7                           | 16.7           | 0.731         |
| Ru        | 48.3                | 45.0                           | 6.7            | 0.762         |
| Average   | 47.2                | 45.0                           | 7.8            | 0.779         |

Table 13: Performance of base model, SFT, and RL tuning models on MGSM. Base denotes the original Qwen2.5-7B-Base model.

| Models           | En     | Zh     | De     | Es     | Fr     | Ja     | Ru     | Th     | Sw    | Bn     | Avg    | Gen    |
|------------------|--------|--------|--------|--------|--------|--------|--------|--------|-------|--------|--------|--------|
| Base             | 75.9   | 50.9   | 56.6   | 68.2   | 62.9   | 50.2   | 64.5   | 48.1   | 4.5   | 39.7   | 52.2   | 0.0    |
| SFT (En)         | 71.8   | 63.5   | 62.5   | 66.3   | 63.3   | 51.1   | 68.3   | 58.4   | 13.1  | 45.2   | 56.4   | 6.8    |
| RL (En)          | 90.9   | 82.2   | 84.1   | 84.9   | 82.1   | 73.8   | 85.1   | 79.3   | 19.2  | 68.3   | 75.0   | 52.2   |
| Δ (RL-SFT)       | +19.1  | +18.7  | +21.6  | +18.6  | +18.8  | +22.7  | +16.8  | +20.9  | +6.1  | +23.1  | +18.6  | +45.4  |
| SFT (Zh)         | 70.7   | 64.7   | 62.1   | 63.5   | 59.8   | 52.5   | 61.7   | 54.7   | 13.0  | 41.8   | 54.4   | 1.8    |
| RL (Zh)          | 92.7   | 83.9   | 82.0   | 85.1   | 83.7   | 75.1   | 83.9   | 78.1   | 19.3  | 68.1   | 75.2   | 53.0   |
| Δ (RL-SFT)       | +22.0  | +19.2  | +19.9  | +21.6  | +23.9  | +22.6  | +22.2  | +23.4  | +6.3  | +26.3  | +20.8  | +51.2  |
| SFT (De)         | 67.5   | 59.9   | 62.7   | 64.0   | 57.7   | 48.9   | 61.5   | 54.6   | 13.9  | 41.5   | 53.2   | -1.6   |
| RL (De)          | 92.3   | 83.7   | 84.2   | 86.3   | 83.5   | 76.3   | 88.5   | 82.8   | 20.9  | 71.1   | 77.0   | 56.7   |
| Δ (RL-SFT)       | +24.8  | +23.8  | +21.5  | +22.3  | +25.8  | +27.4  | +27.0  | +28.2  | +7.0  | +29.6  | +23.8  | +58.3  |

Table 14: Performance of base model, SFT, and RL tuning models on MGSM. Base denotes the original Qwen2.5-3B-Base model. Mix means using the mixture of English, Chinese and German data to tune the base model.

| Models    | En   | Zh   | De   | Es   | Fr   | Ja   | Ru   | Th   | Sw   | Bn   | Avg  | Gen  |
|-----------|------|------|------|------|------|------|------|------|------|------|------|------|
| Base      | 63.4 | 48.3 | 33.5 | 57.7 | 38.9 | 19.5 | 30.3 | 17.6 | 7.3  | 1.2  | 31.8 | 0.0  |
| SFT (En)  | 64.7 | 54.5 | 50.7 | 56.4 | 56.2 | 36.9 | 55.5 | 44.1 | 6.9  | 26.2 | 45.2 | 18.1 |
| RL (En)   | 85.8 | 72.1 | 70.8 | 77.3 | 76.6 | 61.2 | 64.9 | 61.0 | 9.5  | 47.5 | 62.7 | 49.1 |
| SFT (Zh)  | 65.7 | 58.7 | 48.4 | 55.7 | 56.1 | 43.5 | 56.6 | 45.8 | 7.5  | 30.5 | 46.9 | 20.4 |
| RL (Zh)   | 86.1 | 76.3 | 74.2 | 81.1 | 76.1 | 64.5 | 78.1 | 64.9 | 10.3 | 48.3 | 66.0 | 52.6 |
| SFT (De)  | 63.9 | 54.2 | 57.5 | 55.7 | 52.8 | 39.3 | 55.1 | 47.6 | 8.4  | 28.8 | 46.3 | 19.3 |
| RL (De)   | 91.0 | 77.6 | 80.5 | 82.7 | 80.0 | 67.8 | 81.3 | 75.3 | 15.9 | 63.3 | 71.5 | 60.4 |
| SFT (Mix) | 65.3 | 55.9 | 52.9 | 56.3 | 53.0 | 42.9 | 53.2 | 47.1 | 7.6  | 29.8 | 46.4 | 19.6 |
| RL (Mix)  | 87.9 | 75.4 | 77.1 | 79.0 | 79.3 | 64.1 | 78.2 | 69.7 | 12.6 | 57.7 | 68.1 | 55.2 |

Table 15: The performance of the base model, SFT, RL, and cold-start tuning models on MGSM. "Base" denotes the original Qwen2.5-3B-Base model. "SFT + RL" refers to first fine-tuning the base model with SFT, followed by reinforcement learning (RL) tuning using the same dataset. "SFT (100 steps) + RL" indicates that the base model is first fine-tuned with SFT for 100 steps, and then further tuned with RL.

| Models | En | Zh | De | Es | Fr | Ja | Ru | Th | Sw | Bn | Avg | Gen |
|---|---|---|---|---|---|---|---|---|---|---|---|---|
| Base | 63.4 | 48.3 | 33.5 | 57.7 | 38.9 | 19.5 | 30.3 | 17.6 | 7.3 | 1.2 | 31.8 | 0.0 |
| SFT (En) | 64.7 | 54.5 | 50.7 | 56.4 | 56.2 | 36.9 | 55.5 | 44.1 | 6.9 | 26.2 | 45.2 | 18.1 |
| RL (En) | 85.8 | 72.1 | 70.8 | 77.3 | 76.6 | 61.2 | 64.9 | 61.0 | 9.5 | 47.5 | 62.7 | 49.1 |
| SFT + RL (En) | 63.1 | 55.8 | 56.3 | 57.5 | 42.7 | 53.0 | 45.8 | 7.0 | 30.3 | 46.7 | 19.7 |
| SFT (100 steps) (En) | 59.3 | 50.5 | 44.9 | 50.1 | 47.3 | 28.1 | 48.9 | 37.2 | 6.8 | 20.7 | 39.4 | 8.7 |
| SFT (100 steps) + RL (En) | 49.3 | 58.9 | 37.9 | 23.9 | 20.5 | 51.1 | 23.3 | 21.7 | 6.8 | 33.1 | 32.7 | -5.5 |
| SFT (Zh) | 65.7 | 58.7 | 48.4 | 55.7 | 56.1 | 43.5 | 56.6 | 45.8 | 7.5 | 30.5 | 46.9 | 20.4 |
| RL (Zh) | 86.1 | 76.3 | 74.2 | 81.1 | 76.1 | 64.5 | 78.1 | 64.9 | 10.3 | 48.3 | 66.0 | 52.6 |
| SFT + RL (Zh) | 76.3 | 67.8 | 61.7 | 65.9 | 64.0 | 55.4 | 63.1 | 53.1 | 8.3 | 34.5 | 55.0 | 34.5 |
| SFT (100 steps) (Zh) | 60.5 | 48.9 | 43.3 | 49.3 | 47.2 | 32.8 | 48.8 | 37.0 | 5.8 | 18.2 | 39.2 | 8.4 |
| SFT (100 steps) + RL (Zh) | 84.1 | 70.5 | 70.7 | 74.3 | 70.6 | 62.2 | 72.1 | 61.5 | 10.2 | 45.8 | 62.2 | 46.1 |
| SFT (De) | 63.9 | 54.2 | 57.5 | 55.7 | 52.8 | 39.3 | 55.1 | 47.6 | 8.4 | 28.8 | 46.3 | 19.3 |
| RL (De) | 91.0 | 77.6 | 80.5 | 82.7 | 80.0 | 67.8 | 81.3 | 75.3 | 15.9 | 63.3 | 71.5 | 60.4 |
| SFT + RL (De) | 67.9 | 53.5 | 66.0 | 62.7 | 60.6 | 51.0 | 61.9 | 53.5 | 9.9 | 38.9 | 52.6 | 28.8 |
| SFT (100 steps) (De) | 61.1 | 50.4 | 49.6 | 52.6 | 49.7 | 34.7 | 49.0 | 41.1 | 6.9 | 22.7 | 41.8 | 12.3 |
| SFT (100 steps) + RL (De) | 82.1 | 73.5 | 74.3 | 74.7 | 73.8 | 58.8 | 74.2 | 62.3 | 12.5 | 47.3 | 63.4 | 47.7 |

Table 16: Performance of base model, SFT, and RL tuning models on MGSM in the 4-shots setting. Base denotes the original Qwen2.5-3B-Base model.

| Models | En | Zh | De | Es | Fr | Ja | Ru | Avg | Gen |
|---|---|---|---|---|---|---|---|---|---|
| Base | 69.3 | 58.7 | 26.5 | 54.8 | 38.4 | 34.3 | 41.5 | 46.2 | 0.0 |
| SFT (En) | 64.9 | 52.6 | 51.2 | 53.5 | 53.6 | 37.6 | 53.9 | 52.5 | 7.5 |
| RL (En) | 85.5 | 69.9 | 73.9 | 76.4 | 75.9 | 59.9 | 72.0 | 73.4 | 49.2 |
| Δ (RL-SFT) | +20.6 | +17.3 | +22.7 | +22.9 | +22.3 | +22.3 | +18.1 | +20.9 | +41.7 |
| SFT (Zh) | 58.0 | 58.6 | 47.3 | 54.6 | 51.0 | 41.2 | 51.7 | 51.8 | 5.6 |
| RL (Zh) | 86.8 | 75.0 | 70.6 | 79.8 | 75.5 | 62.1 | 79.3 | 75.6 | 54.1 |
| Δ (RL-SFT) | +28.8 | +16.4 | +23.3 | +25.2 | +24.5 | +20.9 | +27.6 | +23.8 | +48.5 |
| SFT (De) | 64.2 | 56.9 | 54.4 | 54.0 | 48.3 | 38.7 | 52.1 | 52.7 | 8.0 |
| RL (De) | 90.5 | 76.8 | 80.1 | 82.9 | 80.5 | 68.9 | 80.7 | 80.1 | 62.3 |
| Δ (RL-SFT) | +26.3 | +19.9 | +25.7 | +28.9 | +32.2 | +30.2 | +28.5 | +27.4 | +54.3 |

Table 17: Performance of base model, SFT, and RL tuning models on M-ifeval under strict scores. Base denotes the orginal Qwen2.5-3B-Base model.

| Models | En | Es | Fr | Ja | Avg | Gen |
|---|---|---|---|---|---|---|
| Base | 40.1 | 46.0 | 40.6 | 23.9 | 37.6 | 0.0 |
| SFT (En) | 34.2 | 44.5 | 40.9 | 21.2 | 35.2 | -3.9 |
| RL (En) | 40.1 | 46.0 | 41.5 | 23.5 | 37.7 | 0.2 |
| Δ (RL-SFT) | +5.9 | +1.5 | +0.6 | +2.2 | +2.5 | +4.1 |
| SFT (Zh) | 36.9 | 40.9 | 35.7 | 21.2 | 33.7 | -6.6 |
| RL (Zh) | 40.8 | 44.5 | 43.2 | 31.0 | 39.9 | 3.0 |
| Δ (RL-SFT) | +3.8 | +3.6 | +7.5 | +9.7 | +6.2 | +9.7 |
| SFT (De) | 32.7 | 37.2 | 29.0 | 23.5 | 30.6 | -12.1 |
| RL (De) | 41.4 | 39.4 | 44.9 | 29.7 | 38.8 | 1.2 |
| Δ (RL-SFT) | +8.6 | +2.2 | +15.9 | +6.2 | +8.2 | +13.4 |

Table 18: Performance of base, SFT, and RL models on multilingual LogiQA. Language codes: En = English, Zh = Chinese, Es = Spanish, Vi = Vietnamese, Id = Indonesian, Ms = Malay, Fil = Filipino.

| Model | En | Zh | Es | Fil | Id | Ms | Vi | Avg | Gen |
|---|---|---|---|---|---|---|---|---|---|
| Base | 35.2 | 27.8 | 35.2 | 1.1 | 3.4 | 4.0 | 15.9 | 17.5 | 0.00 |
| SFT (En) | 42.0 | 38.6 | 35.2 | 24.4 | 38.6 | 31.3 | 34.1 | 34.9 | 19.4 |
| RL (En) | 48.9 | 52.3 | 42.0 | 35.2 | 47.2 | 38.6 | 42.6 | 43.8 | 30.4 |
| $\Delta$ (RL-SFT) | +6.8 | +13.6 | +6.8 | +10.8 | +8.5 | +7.4 | +8.5 | +8.9 | +11.1 |
| SFT (Zh) | 47.7 | 43.8 | 43.8 | 28.4 | 44.9 | 37.5 | 23.3 | 38.5 | 24.1 |
| RL (Zh) | 55.1 | 59.1 | 46.0 | 31.3 | 41.5 | 39.8 | 44.3 | 45.3 | 33.1 |
| $\Delta$ (RL-SFT) | +7.4 | +15.3 | +2.3 | +2.8 | -3.4 | +2.3 | +21.0 | +6.8 | +9.0 |
| SFT (De) | 45.5 | 27.8 | 35.2 | 14.2 | 21.6 | 25.6 | 11.4 | 25.9 | 9.3 |
| RL (De) | 52.3 | 61.4 | 44.3 | 35.8 | 44.9 | 46.0 | 48.3 | 47.6 | 35.3 |
| $\Delta$ (RL-SFT) | +6.8 | +33.5 | +9.1 | +21.6 | +23.3 | +20.5 | +36.9 | +21.7 | +26.0 |

Table 19: Language usage scores on MMath500. We randomly sample 100 questions per language from MMath500 and analyze 6 sampled responses for each question. The scores are calculated based on the rank of language usage within the responses. We report results for the primary languages (En, Zh, De, Es, Fr, Ja) across different models.

| Model | En | Zh | De | Es | Fr | Ja |
|---|---|---|---|---|---|---|
| Base | 0.228 | 0.163 | 0.160 | 0.161 | 0.141 | 0.190 |
| SFT (En) | 0.668 | 0.175 | 0.001 | 0.000 | 0.000 | 0.297 |
| RL (En) | 0.294 | 0.230 | 0.161 | 0.159 | 0.153 | 0.196 |
| SFT (Zh) | 0.309 | 0.420 | 0.077 | 0.092 | 0.097 | 0.350 |
| RL (Zh) | 0.873 | 0.090 | 0.040 | 0.015 | 0.076 | 0.020 |
| SFT (De) | 0.533 | 0.150 | 0.336 | 0.068 | 0.062 | 0.234 |
| RL (De) | 0.819 | 0.112 | 0.073 | 0.010 | 0.021 | 0.096 |

## A.5 CASE STUDY

Table 20: Case study on MMath500. We select the German question to show the case.

| |
|---|
| **Question:** Ein Matheclub veranstaltet einen Kuchenverkauf als Spendenaktion, um Geld für eine bevorstehende Reise zu sammeln. Sie verkaufen 54 Kekse zu drei für \$1\$, und \$20\$ Cupcakes zu 2\$ pro Stück, und \$35\$ Brownies zu \$1\$ pro Stück. Wenn es den Matheclub \$15\$ gekostet hat, diese Artikel zu backen, wie hoch war ihr Gewinn? |
| **Answer**: 78 |

**Base:**

Da Kekse zu drei für \$1 verkaufen werden, müssen wir zahlenweise Kekse zählen. Es gibt 54 Kekse zu backen, und jeder Kuchen besteht aus drei Keksen. Also haben sie $54/3 = 18$ Kuchen zu backen. Zudem haben sie 20 Cupcakes zu backen und 35 Brownies zu backen. Die Matheclub hat also insgesamt $18 + 20 + 35 = 73$ Artikel zu backen. Sind sie zu backen, hat es ihnen $73 \times 2 = \$146$ gekostet. Da sie für die Backzeit 15 Dollar ausgeben, haben sie für den Verkauf $146 - 15 = \$131$ eingenommen. Daher haben sie einen Gewinn von $\boxed{131}$.

**Reinforcement Learning (with German Data):**

1. Calculate the revenue from Kekse: - They sold 54 Kekse at a rate of 3 Kekse for \$1. - The number of sets of 3 Kekse in 54 is $\frac{54}{3} = 18$. - Revenue from Kekse $= 18 \times 1 = 18$ dollars.
2. Calculate the revenue from Cupcakes: - They sold 20 Cupcakes at \$2 each. - Revenue from Cupcakes $= 20 \times 2 = 40$ dollars.
3. Calculate the revenue from Brownies: - They sold 35 Brownies at \$1 each. - Revenue from Brownies $= 35 \times 1 = 35$ dollars.
4. Calculate the total revenue: - Total revenue = Revenue from Kekse + Revenue from Cupcakes + Revenue from Brownies - Total revenue $= 18 + 40 + 35 = 93$ dollars.
5. Calculate the profit: - Cost to make the items = \$15. - Profit = Total revenue - Cost to make the items - Profit $= 93 - 15 = 78$ dollars.
Therefore, the profit is $\boxed{78}$.

**Reinforcement Learning (with German Data and with Consistency Reward):**

Zuerst berechnen wir den Einnahmen von den Keksen. Da sie drei Kekse für \$1 verkaufen, erhalten sie: $54 \div 3 = 18$ Doppelsätze, $18 \times 2 = 36$ Dollar Für die Cupcakes erhalten sie: $20 \div 2 = 10$ Stück, $10 \times 2 = 20$ Dollar Für die Brownies erhalten sie: $35 \div 1 = 35$ Stück, $35 \times 1 = 35$ Dollar Die Gesamtsumme ihrer Einnahmen beträgt: $36 + 20 + 35 = 91$ Dollar Da es ihnen \$15 gekostet hat, um die Artikel zu backen, erhalten sie: $91 - 15 = 76$ Dollar Die Gewinnsumme beträgt $\boxed{76}$.

