# OpenReview forum: "Beyond English-Centric Training: How Reinforcement Learning Improves Cross-Lingual Reasoning in LLMs"
_ICLR.cc/2026/Conference — ICLR 2026 Poster_

### Official Review · Reviewer_58H9 · 2025-10-23

**Soundness:** 2
**Presentation:** 3
**Contribution:** 3
**Rating:** 6
**Confidence:** 4

**Summary:**

This paper performs a systematic investigation into cross-lingual reasoning generalisation of reinforcement learning (RL) and supervised fine-tuning (SFT). Experimental results show that (1) models trained with RL outperform better than SFT and (2) using non-English data achieves better overall performance than using English data.

**Strengths:**

- This paper presents a systematic investigation of the difference between RL and SFT in cross-lingual reasoning generalisation
- This paper shows that using non-English data for RL can more effectively enhance performance and cross-lingual generalisation, which might provide important insights and practice into future multilingual reasoning works
- The authors perform extensive experiments and analysis on several reasoning tasks

**Weaknesses:**

- The training dataset size (GSM8K with around 8k samples) is pretty small and simple, which might influence the findings and conlcusions
- Some details of the experimental setup are missing.
(1)  Inference setting of the base model. It is unclear if the authors use zero-shot or few-shot prompting. Since this is a base model without instruction-tuning, it would be more fair to include few-shot prompting as a baseline;
(2) Training setting of SFT and RL. The authors do not report the training epoch and stopping setting. Compared to the base model, the improvement of SFT is very limited in some languages, such as English, which is unexpected?
- As the results indicate that RL-tuned models exhibit a language-mixing pattern in their outputs, it would be interesting to include statistics on language usage in addition to the language (in)consistency analysis already presented in the paper. In other words, this can show which language the model prefers to use to think and reason, which could be used to analyse cross-lingual transfer and show why RL training on non-English data yields better performance


Others:
- Lines 231-239: The results discussed here are not clearly linked to any table or figure. Is it Table 2?

**Questions:**

See Weaknesses.

---

> ### Author Response · Authors · 2025-11-21
>
> Thank you for your thoughtful and constructive feedback. We sincerely appreciate your recognition of our work and your valuable suggestions for improvement.
>
> ### **About Training Dataset Size** ###
>
> We sincerely appreciate the insight regarding the size of the GSM8K dataset.
>
> Regarding data scale, Table 1 in the paper presents the results of training on MGSM8K (8K samples per language) and testing on MGSM, whereas Table 2 displays the results of training on LUFFY (45K samples per language) and testing on MMath500. It can be observed that as the training data size scales from 8K to 45K, the findings and conclusions that "RL > SFT" and "Non-English RL > English RL" remains consistent.

---

> ### Author Response · Authors · 2025-11-21
>
> ### **About Language Usage Statistics and Analysis**
>
> We sincerely appreciate this highly inspiring suggestion. Including statistics on language usage indeed provides critical evidence for explaining "why RL training on non-English data yields better performance."
>
> We conducted a detailed language analysis on MMath500 by randomly sampling 100 questions for each language (En, Zh, De, Es, Fr, Ja). For each question, we sampled 6 responses and calculated the rank score of language usage to analyze the distribution. We report the scores for the primary languages (En, Zh, De, Es, Fr, Ja) rank scores across different models.
>
> Specifically, for each response, we used Deepseek-v3.2 to annotate the primary languages used, identifying up to 3 languages per response. The scoring rules are as follows:
>
> - Rank 1 language receives a score of 1;
> - Rank 2 language receives a score of 1/2;
> - Rank 3 language receives a score of 1/3;
> - If only one primary language is identified, it receives a score of 1.
>
> We calculated the mean score for each response across different languages, and the results (also  shown in Appendix A.4, Table 19) are:
>
> | Model    | En    | Zh    | De    | Es    | Fr    | Ja    |
> | -------- | ----- | ----- | ----- | ----- | ----- | ----- |
> | Base     | 0.228 | 0.163 | 0.160 | 0.161 | 0.141 | 0.190 |
> | SFT (En) | 0.668 | 0.175 | 0.001 | 0.000 | 0.000 | 0.297 |
> | RL (En)  | 0.294 | 0.230 | 0.161 | 0.159 | 0.153 | 0.196 |
> | SFT (Zh) | 0.309 | 0.420 | 0.077 | 0.092 | 0.097 | 0.350 |
> | RL (Zh)  | 0.873 | 0.090 | 0.040 | 0.015 | 0.076 | 0.020 |
> | SFT (De) | 0.533 | 0.150 | 0.336 | 0.068 | 0.062 | 0.234 |
> | RL (De)  | 0.819 | 0.112 | 0.073 | 0.010 | 0.021 | 0.096 |
>
> The results reveal the following key findings:
>
> 1. **Dominance of English:** The Base Model's language usage is relatively balanced. After SFT and RL training, the usage of English significantly increases. RL-trained models (especially RL-Zh and RL-De) show a significant increase in the usage of English (or English-mixed language) when answering non-English questions. For instance, RL (De) achieves an English usage score of 0.819, far exceeding SFT (De)'s 0.533.
>
> 2. **Adaptive Selection:** The RL training process enables the model to adaptively learn and select suitable languages for complex reasoning, rather than passively adhering to the input language. With the Inconsistency Analysis presented in our paper, this diversity and adaptive selection capability confer better performance to RL, facilitating more effective Cross-Lingual Transfer. In the revised paper, we have added this analysis of language usage analysis in Appendix A.4.1 and the results in Appendix A.4, Table 19.
> 2. **Uncertainty-Driven Exploration:** Prior studies mentioned that Perplexity (PPL) plays a critical role in Reinforcement Learning (RL) [1,2]. To further investigate the underlying causes of the superior performance of RL training on non-English data (especially German data), we employed the base model to generate six responses for each question across different languages via sampling. We then calculated the Perplexity (PPL) of these responses. Additionally, we computed the BLEU score between the multiple sampled responses for each question to serve as a metric for Self-Similarity. The results (also shown in Appendix A.4, Table 11) are:
>
> |                 | En    | Zh    | De    | Fr    | Ja    | Es    |
> | --------------- | ----- | ----- | ----- | ----- | ----- | ----- |
> | Perplexity      | 1.186 | 1.248 | 1.414 | 1.332 | 1.440 | 1.267 |
> | Self-Similarity | 0.621 | 0.505 | 0.425 | 0.448 | 0.433 | 0.534 |
>
> We found that German questions exhibit high PPL (1.414) and the lowest Self-Similarity (0.425 for De vs. 0.621 for En). This indicates that the model faces higher "uncertainty" when processing German questions. In the exploration phase of RL, this uncertainty potentially prompts the model to step out of single-language constraints and attempt reasoning using mixed languages or its strongest language (English), thereby inadvertently activating stronger cross-lingual generalization capabilities. Conversely, responses to English questions exhibit the lowest Perplexity and the highest Self-Similarity, indicating relatively limited diversity in the sampled responses during RL training on English tasks. Furthermore, regarding the Qwen2.5-3B-Base model, although the official pre-training corpus distribution is not explicitly disclosed, English constitutes the majority, followed by a significant portion of Chinese, while other languages are less than English and Chinese (e.g. German). This explains the higher Perplexity for German responses. As a less dominant language in the pre-training data, German induces more diverse sampling during the RL process, which in turn leads to superior cross-lingual generalization after RL training. In the revised paper, we have added this analysis regarding Perplexity and Self-Similarity to Section 4.2 and Table 11 in Appendix A.4.

---

> ### Author Response · Authors · 2025-11-21
>
> ### **About Experimental Settings and Baselines**
>
> Thank you very much for pointing out the missing details in our experimental setup. These suggestions are crucial for enhancing the rigor and reproducibility of our paper.
>
> **(1) Base Model Inference Setting** Our base model indeed utilized a Zero-shot setting. To ensure a fairer comparison, we have also added 4-shots experiment. The results (also shown in Appendix A.4, Table 16) are:
>
> | Models            | En    | Zh    | De    | Es    | Fr    | Ja    | Ru    | Avg   | Gen   |
> | ----------------- | ----- | ----- | ----- | ----- | ----- | ----- | ----- | ----- | ----- |
> | Base              | 69.3  | 58.7  | 26.5  | 54.8  | 38.4  | 34.3  | 41.5  | 46.2  | 0.0   |
> | SFT (En)          | 64.9  | 52.6  | 51.2  | 53.5  | 53.6  | 37.6  | 53.9  | 52.5  | 7.5   |
> | RL (En)           | 85.5  | 69.9  | 73.9  | 76.4  | 75.9  | 59.9  | 72.0  | 73.4  | 49.2  |
> | $\Delta$ (RL-SFT) | +20.6 | +17.3 | +22.7 | +22.9 | +22.3 | +22.3 | +18.1 | +20.9 | +41.7 |
> | SFT (Zh)          | 58.0  | 58.6  | 47.3  | 54.6  | 51.0  | 41.2  | 51.7  | 51.8  | 5.6   |
> | RL (Zh)           | 86.8  | 75.0  | 70.6  | 79.8  | 75.5  | 62.1  | 79.3  | 75.6  | 54.1  |
> | $\Delta$ (RL-SFT) | +28.8 | +16.4 | +23.3 | +25.2 | +24.5 | +20.9 | +27.6 | +23.8 | +48.5 |
> | SFT (De)          | 64.2  | 56.9  | 54.4  | 54.0  | 48.3  | 38.7  | 52.1  | 52.7  | 8.0   |
> | RL (De)           | 90.5  | 76.8  | 80.1  | 82.9  | 80.5  | 68.9  | 80.7  | 80.1  | 62.3  |
> | $\Delta$ (RL-SFT) | +26.3 | +19.9 | +25.7 | +28.9 | +32.2 | +30.2 | +28.5 | +27.4 | +54.3 |
>
> The results demonstrate that the performance trend under the 4-shots setting is highly consistent with the zero-shot setting: RL models still significantly outperform SFT models, and RL (De) continues to exhibit the best cross-lingual generalization (Gen score 62.3). This indicates that our conclusions are robust across different prompting strategies.
>
> **(2) Training Settings for SFT and RL** We have added detailed training setting to the paper in Section 3.1: all SFT and RL experiments were trained for 3 full epochs and then stopped. Regarding the observation that "SFT shows limited improvement over the Base Model on high-resource languages like English," we believe this is not unexpected due to the distinct learning mechanisms:
>
> - **Limitations of SFT:** SFT tends to fit the surface distribution of the training data. For high-resource languages like English, which are already well-learned during pre-training, SFT with samples struggles to provide fundamental improvements in reasoning capabilities and may even limit the model's potential by overfitting to specific formats.
> - **RL Exploration Mechanism:** In contrast, the uncertainty encountered during the Exploration Phase of RL prompts the model to step out of single-language constraints. As we illustrate in "About Language Usage Statistics and Analysis", the higher perplexity (PPL 1.414) and lower self-similarity (0.425) of German questions incentivize the model to attempt reasoning using mixed languages or its strongest language (English). This mechanism inadvertently activates deeper cross-lingual reasoning capabilities, leading to significantly higher performance gains compared to SFT.
>
> ### **About Clarification on Table Reference**
>
> We sincerely appreciate your meticulous suggestion. The results discussed in lines 231-239 (previous version, now in lines 239~247) actually refer to Table 3. We have clarified this reference in the revised paper to avoid any confusion in Section 3.2.
>
> Thank you again for your valuable feedback, particularly regarding the language usage statistics. We are deeply grateful for your suggestions that help us improve the clarity and comprehensiveness of our work, significantly deepening our analysis of the mechanisms behind non-English RL.
>
> **References**
>
> [1] Perplexity-aware correction for robust alignment with noisy preferences.
>
> [2] Rectify Evaluation Preference: Improving LLMs' Critique on Math Reasoning via Perplexity-aware Reinforcement Learning.

---

> > ### Comment · Reviewer_58H9 · 2025-11-26
> > **Official Comment by Reviewer 58H9**
> >
> > Thanks for your rebuttal and additional experiments. I have raised my Soundness score.

---

> > > ### Author Response · Authors · 2025-11-26
> > >
> > > We sincerely appreciate your recognition and valuable feedback. Thank you for your insightful and thorough suggestions, which help us strengthen our work.

---

### Official Review · Reviewer_xepT · 2025-10-25

**Soundness:** 3
**Presentation:** 3
**Contribution:** 2
**Rating:** 4
**Confidence:** 4

**Summary:**

This paper systematically compared Reinforcement learning (RL) with supervised fine-tuning (SFT) for cross-lingual complex reasoning. Experiments on multilingual mathematical, commonsense, and scientific reasoning benchmarks show that RL yields higher accuracy and stronger cross-lingual generalization than SFT. RL trained on non-English data even generalizes better than RL trained on English data, a pattern not observed in SFT. Mechanistic analyses suggest that RL fosters more robust reasoning strategies.

**Strengths:**

1. The paper is well-written and logically structured, making the content easy to follow.
2. Extensive experiments across multiple multilingual reasoning benchmarks provide solid and convincing results.
3. The analysis of model semantic feature shift offers interpretability and valuable insights into the underlying mechanisms.

**Weaknesses:**

1. It would be valuable to investigate whether the observed phenomena also hold for larger models, such as Qwen2.5-7B. Verifying the trends at scale would make the findings more robust and convincing.

2. Prior studies have shown that RL enhances robust reasoning strategies across domains, but this work focuses mainly on multilingual math reasoning, which limits the generality of the conclusions.

3. The training data are obtained through machine translation from English using Qwen3-30B. Given that Qwen3-30B’s multilingual capabilities, this may significantly affect translation quality, especially for low-resource languages. In fact, MathOctopus [1] has already provided a validated multilingual GSM8K dataset(MGSM8K-Instruct).

4. The paper also introduces RFT as a key baseline to analyze the role of sampling, yet crucial experimental details are missing, including sampling size, selection strategy, and decoding temperature.

[1] Breaking Language Barriers in Multilingual Mathematical Reasoning: Insights and Observations.

**Questions:**

1.The main conclusion is drawn only from experiments on small models (3B). How do you ensure that this phenomenon does not simply arise from limited model capacity and implicit English-centric priors?

2.How do you ensure data translation quality, particularly for low-resource languages?

3.What is the complete implementation setup of RFT, including sampling configuration, filtering criteria?

---

> ### Author Response · Authors · 2025-11-21
>
> We sincerely appreciate your constructive and thoughtful feedback, and we hope our reply will solve your confusion:
>
> ### **About Scale Limitation** ###
>
> We sincerely appreciate the suggestion to investigate the phenomena on larger models. We added Qwen2.5-7B-Base as the base model. The results (also reported in Appendix A.4, Table 13) are:
>
> | Models            | En    | Zh    | De    | Es    | Fr    | Ja    | Ru    | Th    | Sw   | Bn    | Avg   | Gen   |
> | ----------------- | ----- | ----- | ----- | ----- | ----- | ----- | ----- | ----- | ---- | ----- | ----- | ----- |
> | Base              | 75.9  | 50.9  | 56.6  | 68.2  | 62.9  | 50.2  | 64.5  | 48.1  | 4.5  | 39.7  | 52.2  | 0.0   |
> | SFT (En)          | 71.8  | 63.5  | 62.5  | 66.3  | 63.3  | 51.1  | 68.3  | 58.4  | 13.1 | 45.2  | 56.4  | 6.8   |
> | RL (En)           | 90.9  | 82.2  | 84.1  | 84.9  | 82.1  | 73.8  | 85.1  | 79.3  | 19.2 | 68.3  | 75.0  | 52.2  |
> | $\Delta$ (RL-SFT) | +19.1 | +18.7 | +21.6 | +18.6 | +18.8 | +22.7 | +16.8 | +20.9 | +6.1 | +23.1 | +18.6 | +45.4 |
> | SFT (Zh)          | 70.7  | 64.7  | 62.1  | 63.5  | 59.8  | 52.5  | 61.7  | 54.7  | 13.0 | 41.8  | 54.4  | 1.8   |
> | RL (Zh)           | 92.7  | 83.9  | 82.0  | 85.1  | 83.7  | 75.1  | 83.9  | 78.1  | 19.3 | 68.1  | 75.2  | 53.0  |
> | $\Delta$ (RL-SFT) | +22.0 | +19.2 | +19.9 | +21.6 | +23.9 | +22.6 | +22.2 | +23.4 | +6.3 | +26.3 | +20.8 | +51.2 |
> | SFT (De)          | 67.5  | 59.9  | 62.7  | 64.0  | 57.7  | 48.9  | 61.5  | 54.6  | 13.9 | 41.5  | 53.2  | -1.6  |
> | RL (De)           | 92.3  | 83.7  | 84.2  | 86.3  | 83.5  | 76.3  | 88.5  | 82.8  | 20.9 | 71.1  | 77.0  | 56.7  |
> | $\Delta$ (RL-SFT) | +24.8 | +23.8 | +21.5 | +22.3 | +25.8 | +27.4 | +27.0 | +28.2 | +7.0 | +29.6 | +23.8 | +58.3 |
>
> Results indicate that the 7B model exhibits a trend highly consistent with the 3B model: (1) RL achieves significantly higher performance gains compared to SFT. (2) RL training on non-English data (e.g., German, Chinese) continues to demonstrate stronger cross-lingual generalization capabilities than RL on English data (Average 77.0 for RL (De)  vs 75.0 for RL (En)). This provides compelling evidence that the "Non-English RL > English RL" effect is not specific to small models but holds as a general law at larger parameter scales. In the revised paper, we have reported the full results of the Qwen2.5-7B-Base model in Appendix A.4, Table 13, confirming the scalability of our findings.
>
> Furthermore, regarding the concern about implicit English-centric priors, we carefully considered the characteristics of the base models. While most current foundation models are predominantly trained on English corpora, the Qwen2.5 series (Qwen2.5-3B-Base and Qwen2.5-7B-Base) we selected incorporates a significantly higher proportion of Chinese data during pre-training compared to other open-source models. Consequently, the influence of implicit English-centric priors is relatively weaker in our experimental setup. The fact that the advantages of non-English RL remain significant even on these more linguistically balanced models further validates the robustness of our findings.

---

> ### Author Response · Authors · 2025-11-21
>
> ### **Generalization across Domains and Tasks** ###
>
> We appreciate the suggestion regarding the domain coverage of our experiments.  In the original paper, we presented performance on multilingual mathematical reasoning (MGSM and MMath500) and knowledge-oriented reasoning tasks (MMLU-ProX-Lite and MGPQA-D). To further demonstrate the universality of our conclusions,  we have added the instruction-following task M-ifEval [1], and the Multilingual LogiQA [2] task which emphasizes logical capabilities.
>
> The results (also reported in Appendix A.4, Table 17) of M-ifEval are:
>
> | Models            | En   | Es   | Fr    | Ja   | Avg  | Gen   |
> | ----------------- | ---- | ---- | ----- | ---- | ---- | ----- |
> | Base              | 40.1 | 46.0 | 40.6  | 23.9 | 37.6 | 0.0   |
> | SFT (En)          | 34.2 | 44.5 | 40.9  | 21.2 | 35.2 | -3.9  |
> | RL (En)           | 40.1 | 46.0 | 41.5  | 23.5 | 37.7 | 0.2   |
> | $\Delta$ (RL-SFT) | +5.9 | +1.5 | +0.6  | +2.2 | +2.5 | +4.1  |
> | SFT (Zh)          | 36.9 | 40.9 | 35.7  | 21.2 | 33.7 | -6.6  |
> | RL (Zh)           | 40.8 | 44.5 | 43.2  | 31.0 | 39.9 | 3.0   |
> | $\Delta$ (RL-SFT) | +3.8 | +3.6 | +7.5  | +9.7 | +6.2 | +9.7  |
> | SFT (De)          | 32.7 | 37.2 | 29.0  | 23.5 | 30.6 | -12.1 |
> | RL (De)           | 41.4 | 39.4 | 44.9  | 29.7 | 38.8 | 1.2   |
> | $\Delta$ (RL-SFT) | +8.6 | +2.2 | +15.9 | +6.2 | +8.2 | +13.4 |
>
> The results (also reported in Appendix A.4, Table 18) of Multilingual LogiQA are:
>
> | Model             | En   | Zh    | Es   | Fil   | Id    | Ms    | Vi    | Avg   | Gen   |
> | ----------------- | ---- | ----- | ---- | ----- | ----- | ----- | ----- | ----- | ----- |
> | Base              | 35.2 | 27.8  | 35.2 | 1.1   | 3.4   | 4.0   | 15.9  | 17.5  | 0.00  |
> | SFT (En)          | 42.0 | 38.6  | 35.2 | 24.4  | 38.6  | 31.3  | 34.1  | 34.9  | 19.4  |
> | RL (En)           | 48.9 | 52.3  | 42.0 | 35.2  | 47.2  | 38.6  | 42.6  | 43.8  | 30.4  |
> | $\Delta$ (RL-SFT) | +6.8 | +13.6 | +6.8 | +10.8 | +8.5  | +7.4  | +8.5  | +8.9  | +11.1 |
> | SFT (Zh)          | 47.7 | 43.8  | 43.8 | 28.4  | 44.9  | 37.5  | 23.3  | 38.5  | 24.1  |
> | RL (Zh)           | 55.1 | 59.1  | 46.0 | 31.3  | 41.5  | 39.8  | 44.3  | 45.3  | 33.1  |
> | $\Delta$ (RL-SFT) | +7.4 | +15.3 | +2.3 | +2.8  | -3.4  | +2.3  | +21.0 | +6.8  | +9.0  |
> | SFT (De)          | 45.5 | 27.8  | 35.2 | 14.2  | 21.6  | 25.6  | 11.4  | 25.9  | 9.3   |
> | RL (De)           | 52.3 | 61.4  | 44.3 | 35.8  | 44.9  | 46.0  | 48.3  | 47.6  | 35.3  |
> | $\Delta$ (RL-SFT) | +6.8 | +33.5 | +9.1 | +21.6 | +23.3 | +20.5 | +36.9 | +21.7 | +26.0 |
>
> On M-ifEval, RL (De) achieved a score of 38.8 under strict evaluation, outperforming SFT (De) at 30.6 and RL (En) at 37.7 on the average score. On Multilingual LogiQA, RL (De) also achieved the highest average score (47.6) and generalization score (35.3). These consistent cross-task results indicate that the advantages brought by RL are not limited to mathematical reasoning but also improve the model's general cross-lingual capabilities in instruction following and logical reasoning.  Moreover, it is evident that RL training on non-English data yields superior generalization compared to RL on English data. This advantage extends beyond cross-lingual generalization to demonstrate enhanced cross-task generalization. In the revised paper, we have added these cross-task generalization results on M-ifEval and Multilingual LogiQA to Appendix A.4 (Table 17 and Table 18) and the demonstration in Section 3.3.
>
> **References**
>
> [1] SeaEval for multilingual foundation models: From cross-lingual alignment to cultural reasoning.
>
> [2] M-ifeval: Multilingual instruction-following evaluation.

---

> ### Author Response · Authors · 2025-11-21
>
> ### **About Translation Quality** ###
>
> We deeply appreciate the concern regarding the translation quality of our training data using Qwen3-30B-A3B. To ensure high translation quality, we implemented an automated verification mechanism during the translation process using Deepseek-V3. Instances that failed this verification were regenerated until they met the quality standards.
>
> MathOctopus indeed offers a verified and high-quality multilingual GSM8K dataset, which we also cited in the paper. However, the LUFFY dataset is originally available only in English. To maintain consistent multilingual experimental settings across different training data, we employed Qwen3-30B-A3B to translate both GSM8K and LUFFY, thereby generating uniform multilingual training data.
>
> Furthermore, although MGSM8K-Instruct is a high-quality dataset suitable for multilingual training, its GSM8KInstruct_Parallel subset does not fully satisfy the strict data alignment requirements of our study (where identical data content across languages is necessary). As a result of rule-based extraction, only the first 20 examples are strictly aligned one-to-one across different languages.
>
> To further verify translation quality, we sampled translation examples from each language and used Deepseek-V3.2, Deepseek-R1, and GPT-4o as judges to conduct a head-to-head quality comparison between our translations and the validated MGSM8K-Instruct dataset. We utilized the first 20 strictly aligned examples from MGSM8K-Instruct (20 per language) as a benchmark to conduct a direct comparison with our corresponding translated data. The results (also reported in Appendix A.4, Table 12) are:
>
> | Languages | Ours Wins Ratio (%) | MGSM8K-Instruct Wins Ratio (%) | Ties Ratio (%) | Fleiss Kappa |
> | --------- | ------------------- | ------------------------------ | -------------- | ------------ |
> | Zh        | 63.3                | 33.3                           | 3.3            | 0.726        |
> | De        | 46.7                | 48.3                           | 5.0            | 0.817        |
> | Fr        | 40.0                | 50.0                           | 10.0           | 0.885        |
> | Ja        | 48.3                | 46.7                           | 5.0            | 0.756        |
> | Es        | 36.7                | 46.7                           | 16.7           | 0.731        |
> | Ru        | 48.3                | 45.0                           | 6.7            | 0.762        |
> | Average   | 47.2                | 45.0                           | 7.8            | 0.779        |
>
> Results show that our translation data has a win rate comparable to MGSM8K-Instruct (Average Wins: Ours 47.2% vs MGSM8K 45.0%), demonstrating the high quality of the training data, which is comparable to the high-quality MGSM8K-Instruct dataset.
>
> Additionally, translations into Chinese exhibit relatively higher quality due to the Qwen series' extensive exposure to Chinese corpus during pre-training. While the quality is slightly lower for languages with fewer resources, the comparable overall win rates confirm the validity and utility of our translated dataset.
>
> In the revised paper, we have added this translation quality comparison against MGSM8K-Instruct to Appendix A.3 and the results in Table 12.
>
> ### **About RFT Experimental Details** ###
>
> We sincerely appreciate the reminder to include the missing implementation details for Rejection Sampling Fine-Tuning (RFT). In the RFT experiments, we sampled from the RL-trained model on the training set. The specific parameter settings were: Temperature = 1.0, Top-p = 0.95. For each prompt, we sampled 10 times and filtered for responses with correct answers for fine-tuning. We strictly controlled the RFT training data volume to be consistent with SFT. Aside from the data source, the RFT training hyperparameters were identical to SFT. We have added these details to the revised paper in Appendix A.2.1.
>
> Thank you again for your constructive feedback. We are sincerely grateful for your suggestions that help us improve the clarity and comprehensiveness of our work, enhancing the reproducibility, rigorousness, and robustness of our experiments.

---

> > ### Comment · Reviewer_xepT · 2025-11-23
> >
> > Thank you for your thorough rebuttal. I appreciate the detailed clarifications and the effort to address the points I raised. I will raise my score.

---

> > > ### Author Response · Authors · 2025-11-23
> > >
> > > We deeply appreciate your recognition and thorough review! Thank you for the constructive feedback that strengthens our work.

---

### Official Review · Reviewer_fFzm · 2025-10-30

**Soundness:** 4
**Presentation:** 3
**Contribution:** 2
**Rating:** 4
**Confidence:** 4

**Summary:**

The paper investigates how reinforcement learning (RL) vs. supervised fine-tuning (SFT) affects cross-lingual reasoning when training small LLMs (Qwen2.5-3B-Base, plus a check on SmolLM3-3B-Base). The key findings are: (1) RL substantially outperforms SFT on multilingual benchmarks and transfers better across languages; (2) non-English RL training (notably German, also Chinese/Japanese/French) produces higher average accuracy and generalization than English RL, a pattern not observed for SFT. Mechanistic analyses point to (i) language inconsistency during reasoning (models trained with German instructions frequently reason in mixed/non-German), (ii) benefits of on-policy sampling/exploration beyond SFT/RFT, and (iii) smaller semantic drift from pretraining features correlating with better cross-lingual generalization.

**Strengths:**

1. Intuitive central idea with clear, repeated gains over SFT across languages/datasets.
2. Actionable takeaway for the community: train RL on non-English (esp. German/Chinese) to improve cross-lingual generalization.
3. Solid experimental practice: MGSM/MMath500/MAIME2024 (6×/6×/16×); ~10 languages;
4. Insightful analysis:
- (a) “Trained with German, might not reason strictly in German”—documented via Table 5 examples and Table 6 consistency stats.
- (b) Prompt/consistency reward decreases accuracy (inverse correlation) → linguistic flexibility appears beneficial. Figure 2 & Table 6.
- (c) Sampling/online RL > SFT/RFT for exploration and alignment with the model’s distribution.
- (d) Semantic-shift analysis: smaller representational drift (PCA diffs) aligns with stronger cross-lingual generalization.

**Weaknesses:**

1. Scale limitation: results are shown on two 3B models; it’s unclear if the “non-English RL > English RL” effect holds at a larger model and dataset scale.
2. Domain breadth: math and knowledge-oriented reasoning are well covered; fewer open-ended or instruction-following multilingual tasks.
3. Mechanistic claims are suggestive (correlational): e.g., language inconsistency ↔ performance—great ablations, but causal proof remains preliminary.

**Questions:**

1. Prompt/consistency vs. accuracy inversion (Fig. 2 & Table 6): could you run a crossed intervention where the prompt explicitly requests mixed/other language while the reward penalizes similarity to the input language (an inconsistency reward)—i.e., the opposite of your current setting? This would help stress-test whether encouraged inconsistency further improves RL, or whether the benefit is emergent rather than enforced.
2. Why is German so strong for RL? Please discuss whether advantages come from (a) dataset bias/translation quality in LUFFY/GSM8K translations, (b) pretraining distribution (German coverage/quality), or (c) intrinsic linguistic properties (morphology/compounding influencing tokenization and reasoning decomposability). Any measurements—e.g., token length, subword overlap with English/Chinese, or per-language perplexity/coverage—would help separate these hypotheses.

---

> ### Author Response · Authors · 2025-11-21
>
> We sincerely appreciate your insightful and constructive feedback, and we hope that our responses will clarify your concerns:
>
> ### **About Scale Limitation** ###
>
> We sincerely appreciate the suggestion to verify our findings on larger-scale models to ensure robustness. We added Qwen2.5-7B-Base  as the base model. The results (also reported in Appendix A.4, Table 13) are:
>
> | Models            | En    | Zh    | De    | Es    | Fr    | Ja    | Ru    | Th    | Sw   | Bn    | Avg   | Gen   |
> | ----------------- | ----- | ----- | ----- | ----- | ----- | ----- | ----- | ----- | ---- | ----- | ----- | ----- |
> | Base              | 75.9  | 50.9  | 56.6  | 68.2  | 62.9  | 50.2  | 64.5  | 48.1  | 4.5  | 39.7  | 52.2  | 0.0   |
> | SFT (En)          | 71.8  | 63.5  | 62.5  | 66.3  | 63.3  | 51.1  | 68.3  | 58.4  | 13.1 | 45.2  | 56.4  | 6.8   |
> | RL (En)           | 90.9  | 82.2  | 84.1  | 84.9  | 82.1  | 73.8  | 85.1  | 79.3  | 19.2 | 68.3  | 75.0  | 52.2  |
> | $\Delta$ (RL-SFT) | +19.1 | +18.7 | +21.6 | +18.6 | +18.8 | +22.7 | +16.8 | +20.9 | +6.1 | +23.1 | +18.6 | +45.4 |
> | SFT (Zh)          | 70.7  | 64.7  | 62.1  | 63.5  | 59.8  | 52.5  | 61.7  | 54.7  | 13.0 | 41.8  | 54.4  | 1.8   |
> | RL (Zh)           | 92.7  | 83.9  | 82.0  | 85.1  | 83.7  | 75.1  | 83.9  | 78.1  | 19.3 | 68.1  | 75.2  | 53.0  |
> | $\Delta$ (RL-SFT) | +22.0 | +19.2 | +19.9 | +21.6 | +23.9 | +22.6 | +22.2 | +23.4 | +6.3 | +26.3 | +20.8 | +51.2 |
> | SFT (De)          | 67.5  | 59.9  | 62.7  | 64.0  | 57.7  | 48.9  | 61.5  | 54.6  | 13.9 | 41.5  | 53.2  | -1.6  |
> | RL (De)           | 92.3  | 83.7  | 84.2  | 86.3  | 83.5  | 76.3  | 88.5  | 82.8  | 20.9 | 71.1  | 77.0  | 56.7  |
> | $\Delta$ (RL-SFT) | +24.8 | +23.8 | +21.5 | +22.3 | +25.8 | +27.4 | +27.0 | +28.2 | +7.0 | +29.6 | +23.8 | +58.3 |
>
> Results indicate that the 7B model exhibits a trend highly consistent with the 3B model: (1) RL achieves significantly higher performance gains compared to SFT; (2) RL training on non-English data (e.g., German, Chinese) continues to demonstrate stronger cross-lingual generalization capabilities than RL on English data (Average 77.0 for RL (De)  vs 75.0 for RL (En)). This provides compelling evidence that the "Non-English RL > English RL" effect is not specific to small models but holds as a general law at larger parameter scales. In the revised paper, we have reported the full results of the Qwen2.5-7B-Base model in Appendix A.4, Table 13, confirming the scalability of our findings.
>
> Furthermore, regarding data scale, Table 1 in the paper presents the results of training on MGSM8K (8K samples per language) and testing on MGSM, whereas Table 2 displays the results of training on LUFFY (45K samples per language) and testing on MMath500. It can be observed that as the training data size scales from 8K to 45K, the conclusion that "Non-English RL > English RL" remains consistent.

---

> ### Author Response · Authors · 2025-11-21
>
> ### **Generalization across Domains and Tasks** ###
>
> We appreciate the suggestion regarding the domain coverage of our experiments. To demonstrate the universality of our conclusions, in addition to math and knowledge reasoning, we have added the instruction-following task M-ifEval [1], and the Multilingual LogiQA [2] task which emphasizes logical capabilities.
>
> The results (also reported in Appendix A.4, Table 17) of M-ifEval are:
>
> | Models            | En   | Es   | Fr    | Ja   | Avg  | Gen   |
> | ----------------- | ---- | ---- | ----- | ---- | ---- | ----- |
> | Base              | 40.1 | 46.0 | 40.6  | 23.9 | 37.6 | 0.0   |
> | SFT (En)          | 34.2 | 44.5 | 40.9  | 21.2 | 35.2 | -3.9  |
> | RL (En)           | 40.1 | 46.0 | 41.5  | 23.5 | 37.7 | 0.2   |
> | $\Delta$ (RL-SFT) | +5.9 | +1.5 | +0.6  | +2.2 | +2.5 | +4.1  |
> | SFT (Zh)          | 36.9 | 40.9 | 35.7  | 21.2 | 33.7 | -6.6  |
> | RL (Zh)           | 40.8 | 44.5 | 43.2  | 31.0 | 39.9 | 3.0   |
> | $\Delta$ (RL-SFT) | +3.8 | +3.6 | +7.5  | +9.7 | +6.2 | +9.7  |
> | SFT (De)          | 32.7 | 37.2 | 29.0  | 23.5 | 30.6 | -12.1 |
> | RL (De)           | 41.4 | 39.4 | 44.9  | 29.7 | 38.8 | 1.2   |
> | $\Delta$ (RL-SFT) | +8.6 | +2.2 | +15.9 | +6.2 | +8.2 | +13.4 |
>
> The results (also reported in Appendix A.4, Table 18) of Multilingual LogiQA are:
>
> | Model             | En   | Zh    | Es   | Fil   | Id    | Ms    | Vi    | Avg   | Gen   |
> | ----------------- | ---- | ----- | ---- | ----- | ----- | ----- | ----- | ----- | ----- |
> | Base              | 35.2 | 27.8  | 35.2 | 1.1   | 3.4   | 4.0   | 15.9  | 17.5  | 0.00  |
> | SFT (En)          | 42.0 | 38.6  | 35.2 | 24.4  | 38.6  | 31.3  | 34.1  | 34.9  | 19.4  |
> | RL (En)           | 48.9 | 52.3  | 42.0 | 35.2  | 47.2  | 38.6  | 42.6  | 43.8  | 30.4  |
> | $\Delta$ (RL-SFT) | +6.8 | +13.6 | +6.8 | +10.8 | +8.5  | +7.4  | +8.5  | +8.9  | +11.1 |
> | SFT (Zh)          | 47.7 | 43.8  | 43.8 | 28.4  | 44.9  | 37.5  | 23.3  | 38.5  | 24.1  |
> | RL (Zh)           | 55.1 | 59.1  | 46.0 | 31.3  | 41.5  | 39.8  | 44.3  | 45.3  | 33.1  |
> | $\Delta$ (RL-SFT) | +7.4 | +15.3 | +2.3 | +2.8  | -3.4  | +2.3  | +21.0 | +6.8  | +9.0  |
> | SFT (De)          | 45.5 | 27.8  | 35.2 | 14.2  | 21.6  | 25.6  | 11.4  | 25.9  | 9.3   |
> | RL (De)           | 52.3 | 61.4  | 44.3 | 35.8  | 44.9  | 46.0  | 48.3  | 47.6  | 35.3  |
> | $\Delta$ (RL-SFT) | +6.8 | +33.5 | +9.1 | +21.6 | +23.3 | +20.5 | +36.9 | +21.7 | +26.0 |
>
> On M-ifEval, RL (De) achieved a score of 38.8 under strict evaluation, outperforming SFT (De) at 30.6 and RL (En) at 37.7 on the average score. On Multilingual LogiQA, RL (De) also achieved the highest average score (47.6) and generalization score (35.3). These consistent cross-task results indicate that the advantages brought by RL are not limited to mathematical reasoning but also improve the model's general cross-lingual capabilities in instruction following and logical reasoning.  Moreover, it is evident that RL training on non-English data yields superior generalization compared to RL on English data. This advantage extends beyond cross-lingual generalization to demonstrate enhanced cross-task generalization. In the revised paper, we have added these cross-task generalization results on M-ifEval and Multilingual LogiQA to Appendix A.4 (Table 17 and Table 18) and the demonstration in Section 3.3.
>
> **References**
>
> [1] SeaEval for multilingual foundation models: From cross-lingual alignment to cultural reasoning.
>
> [2] M-ifeval: Multilingual instruction-following evaluation.

---

> ### Author Response · Authors · 2025-11-21
>
> ### **Analysis of the Causes of the Superiority of RL on German Data** ###
>
> We deeply appreciate the insightful question regarding the reasons why RL on German data exhibits a significant advantage.
>
> Prior studies mentioned that Perplexity (PPL) plays a critical role in Reinforcement Learning (RL) [1,2]. To investigate the underlying causes of the superior performance of German data in RL, we employed the base model to generate six responses for each question across different languages via sampling. We then calculated the Perplexity (PPL) of these responses. Additionally, we computed the BLEU score between the multiple sampled responses for each question to serve as a metric for Self-Similarity. The results  (also  shown in Appendix A.4, Table 11) are:
>
> |                 | En    | Zh    | De    | Fr    | Ja    | Es    |
> | --------------- | ----- | ----- | ----- | ----- | ----- | ----- |
> | Perplexity      | 1.186 | 1.248 | 1.414 | 1.332 | 1.440 | 1.267 |
> | Self-Similarity | 0.621 | 0.505 | 0.425 | 0.448 | 0.433 | 0.534 |
>
> We found that German  questions exhibit high PPL (1.414) and the lowest Self-Similarity (0.425 for De vs. 0.621 for En). This indicates that the model faces higher "uncertainty" when processing German questions. In the exploration phase of RL, this uncertainty potentially prompts the model to step out of single-language constraints and attempt reasoning using mixed languages or its strongest language (English), thereby inadvertently activating stronger cross-lingual generalization capabilities. Conversely, responses to English questions exhibit the lowest Perplexity and the highest Self-Similarity, indicating relatively limited diversity in the sampled responses during RL training on English tasks. Furthermore, regarding the Qwen2.5-3B-Base model, although the official pre-training corpus distribution is not explicitly disclosed, English constitutes the majority, followed by a significant portion of Chinese, while other languages are less than English and Chinese (e.g. German). This explains the higher Perplexity for German responses. As a less dominant language in the pre-training data, German induces more diverse sampling during the RL process, which in turn leads to superior cross-lingual generalization after RL training.
>
> Furthermore, in addition to German, both French and Japanese questions elicit relatively high PPL values and low Self-Similarity scores. This aligns with the results reported in Table 7 of the Appendix in the paper, which demonstrate that using Japanese and French data for RL training also yields superior performance and generalization capabilities.
>
> In the revised paper, we have added this analysis regarding Perplexity and Self-Similarity to Section 4.2 and Table 11 in Appendix A.4.
>
> We are deeply grateful for this insightful comment, as it inspired us to delve deeper into the mechanisms behind the RL's superiority. These results not only clarify the current phenomenon but also open up exciting avenues for our future research in enhancing multilingual capabilities.
>
> **References**
>
> [1] Perplexity-aware correction for robust alignment with noisy preferences.
>
> [2] Rectify Evaluation Preference: Improving LLMs' Critique on Math Reasoning via Perplexity-aware Reinforcement Learning.

---

> ### Author Response · Authors · 2025-11-21
>
> ### **Crossed Intervention Performance and Consistency Causal Analysis** ###
>
> We sincerely appreciate the suggestion to conduct a crossed intervention to stress-test the causal relationship between language inconsistency and performance. We designed a "Reverse Intervention" experiment: explicitly encouraging inconsistent language in the Prompt and rewarding inconsistent behavior in the Reward (Inconsistency Prompt and Reward).
>
> The results (also reported in Appendix A.4, Table 9) are:
>
> | **Models**                        | **Zh** | **Fr** | **En** | **De** | **Ja** | **Es** | **Average** |
> | --------------------------------- | ------ | ------ | ------ | ------ | ------ | ------ | ----------- |
> | Base                              | 38.9   | 27.2   | 49.1   | 16.3   | 17.4   | 36.6   | 30.9        |
> | RL (En)                           | 53.7   | 55.8   | 62.7   | 50.9   | 54.2   | 56.6   | 55.7        |
> | RL (Zh)                           | 61.3   | 61.2   | 63.3   | 61.2   | 58.5   | 62.1   | 61.3        |
> | + Consistency Prompt              | 53.3   | 54.9   | 59.7   | 42.2   | 55.1   | 56.8   | 53.7        |
> | + Consistency Prompt and Reward   | 56.2   | 54.2   | 62.9   | 45.5   | 48.2   | 56.4   | 53.9        |
> | + Inconsistency Prompt and Reward | 59.9   | 56.1   | 61.6   | 57.6   | 57.7   | 58.2   | 58.5        |
> | RL (De)                           | 61.4   | 61.5   | 62.8   | 60.7   | 60.1   | 62.1   | 61.4        |
> | + Consistency Prompt              | 56.0   | 61.3   | 63.8   | 60.8   | 59.0   | 61.9   | 60.5        |
> | + Consistency Prompt and Reward   | 51.9   | 52.1   | 62.4   | 49.3   | 41.6   | 54.6   | 52.0        |
> | + Inconsistency Prompt and Reward | 59.1   | 60.1   | 62.4   | 60.1   | 57.5   | 60.9   | 60.0        |
>
> The experiment reveals that while encouraging "inconsistency" yields better performance than enforcing "consistency" (validating that inconsistency is beneficial), allowing the RL model to autonomously select the language (i.e., the original RL setting without language constraints) still achieves the best results.
>
> This indicates that "language inconsistency" is a key phenomenon and factor in RL's enhanced generalization. However, such language inconsistency is merely one contributing factor to the high performance and generalization of RL. Furthermore, as discussed in "Analysis of the Causes of the Superiority of RL on German Data", the advantage of RL also stems from its free exploration of reasoning paths. Forced interventions (whether consistent or inconsistent) inevitably restrict this exploration space to some extent.
>
> In the revised paper, the results were updated in Figure 2, Table 6 and Table 9 and we have incorporated this analysis of inconsistency in Section 4.1.
>
> Thank you again for your valuable feedback. We deeply appreciate your constructive suggestions to delve deeper into the non-English RL phenomenon. Your suggestions to verify findings on larger-scale models and analyze the causes of superiority has made our conclusions regarding the "Non-English RL advantage" much more robust and convincing.

---

### Official Review · Reviewer_hEuo · 2025-11-01

**Soundness:** 3
**Presentation:** 4
**Contribution:** 2
**Rating:** 6
**Confidence:** 4

**Summary:**

This paper investigates if RL could be beneficial for cross-lingual generalization compare to that of SFT. The authors train Qwen2.5-3B-Base on translated GSM8K and LUFFY datasets (into Chinese and German) and ablate training on SFT and RL. The models are evaluated across ten languages on 4 benchmarks (MGSM, MMath500, MMLU-ProX-Lite, and MGPQA-Diamond), where RL consistently outperforms SFT across different settings.

**Strengths:**

1. This paper addresses an important question of how RL training affects cross-lingual generalization, and the finding that RL shows better generalization is very insightful for practitioners.

2. The paper is written very clearly, and it was easy to follow the content while reading.

3. The findings in Section 4.2., in that online optimization is also beneficial for cross-lingual generalization (in addition to "reasoning", which the community has most focused on) highlights the benefit of how the learning algorithms could affect generalization in a broader scope. The analysis shown in Figure 4 can also be adopted in future work.

**Weaknesses:**

1. Across Table 1,2,3,4, the performance of base model, SFT'ed model, and RL'ed model are shown. In practice, a lot of people also first apply SFT (referred to as cold-start) and then apply RL on top of that. Would the performance of such baseline outperform the only RL'ed variant or not? I think such cold-start is adopted a lot in practice, so there should be a baseline added.

2. Taking the French scores on MGSM (Table 1) for example, when training with SFT, the Chinese dataset yields higher performance than German dataset (56.1 vs 52.8). However, with RL, the German dataset yields higher performance than Chinese dataset (80.0 vs 76.1). What could be the reason why the transferability differs across training algorithms and how does that relate to the similarity between the language used for training and the target language that is evaluated? Although it might be difficult to answer this question holistically, I think since the motivation of this work is on cross-lingual generalization w.r.t different algorithms, there should be a minimal explanation related to this.

3. How would the performance differ when applying SFT or RL with English, Chinese, and German datasets combined? I think this is the typical set-up done for improving multilingual capabilities of LMs, hence it should be added to the results section. Does the performance drop compared to that of training on a single target language?

4. How is the language consistency reward defined in Equation 1, and how is it implmeneted/measured? I think there should be more explanation of this in the text body.

**Questions:**

See weaknesses above

---

> ### Author Response · Authors · 2025-11-21
>
> Thank you for your recognition and constructive feedback, and we hope that our reply can solve your confusion.
>
> ### **About Cold-Start Baseline** ###
>
> We sincerely appreciate the importance of the cold-start baseline, as it is indeed a widely adopted practice. We have added "Cold-Start" experiments. To provide a comprehensive analysis, we adopted two settings:
>
> 1. **SFT + RL**: Fully fine-tuning the base model with SFT (identical to the SFT setting in Table 1), followed by RL tuning.
> 2. **SFT (100 steps) + RL**: Fine-tuning the base model with SFT for only 100 steps as a warm-up, followed by RL.
>
> The results (also shown in Appendix A.4, Table 15) are:
>
> | Models                     | En   | Zh   | De   | Es   | Fr   | Ja   | Ru   | Th   | Sw   | Bn   | Avg  | Gen  |
> | -------------------------- | ---- | ---- | ---- | ---- | ---- | ---- | ---- | ---- | ---- | ---- | ---- | ---- |
> | Base                       | 63.4 | 48.3 | 33.5 | 57.7 | 38.9 | 19.5 | 30.3 | 17.6 | 7.3  | 1.2  | 31.8 | 0.0  |
> | SFT (En)                   | 64.7 | 54.5 | 50.7 | 56.4 | 56.2 | 36.9 | 55.5 | 44.1 | 6.9  | 26.2 | 45.2 | 18.1 |
> | RL (En)                    | 85.8 | 72.1 | 70.8 | 77.3 | 76.6 | 61.2 | 64.9 | 61.0 | 9.5  | 47.5 | 62.7 | 49.1 |
> | SFT + RL (En)              | 63.1 | 55.8 | 56.3 | 55.3 | 57.5 | 42.7 | 53.0 | 45.8 | 7.0  | 30.3 | 46.7 | 19.7 |
> | SFT (100 steps) (En)       | 59.3 | 50.5 | 44.9 | 50.1 | 47.3 | 28.1 | 48.9 | 37.2 | 6.8  | 20.7 | 39.4 | 8.7  |
> | SFT (100 steps) + RL (En)  | 49.3 | 58.9 | 37.9 | 23.9 | 20.5 | 51.1 | 23.3 | 21.7 | 6.8  | 33.1 | 32.7 | -5.5 |
> | SFT (Zh)                   | 65.7 | 58.7 | 48.4 | 55.7 | 56.1 | 43.5 | 56.6 | 45.8 | 7.5  | 30.5 | 46.9 | 20.4 |
> | RL (Zh)                    | 86.1 | 76.3 | 74.2 | 81.1 | 76.1 | 64.5 | 78.1 | 64.9 | 10.3 | 48.3 | 66.0 | 52.6 |
> | SFT+RL (Zh)                | 76.3 | 67.8 | 61.7 | 65.9 | 64.0 | 55.4 | 63.1 | 53.1 | 8.3  | 34.5 | 55.0 | 34.5 |
> | SFT (100 steps) (Zh)       | 60.5 | 48.9 | 43.3 | 49.3 | 47.2 | 32.8 | 48.8 | 37.0 | 5.8  | 18.2 | 39.2 | 8.4  |
> | SFT (100 steps)+RL (Zh)    | 84.1 | 70.5 | 70.7 | 74.3 | 70.6 | 62.2 | 72.1 | 61.5 | 10.2 | 45.8 | 62.2 | 46.1 |
> | SFT (De)                   | 63.9 | 54.2 | 57.5 | 55.7 | 52.8 | 39.3 | 55.1 | 47.6 | 8.4  | 28.8 | 46.3 | 19.3 |
> | RL (De)                    | 91.0 | 77.6 | 80.5 | 82.7 | 80.0 | 67.8 | 81.3 | 75.3 | 15.9 | 63.3 | 71.5 | 60.4 |
> | SFT + RL (De)              | 67.9 | 53.5 | 66.0 | 62.7 | 60.6 | 51.0 | 61.9 | 53.5 | 9.9  | 38.9 | 52.6 | 28.8 |
> | SFT (100 steps) (De)       | 61.1 | 50.4 | 49.6 | 52.6 | 49.7 | 34.7 | 49.0 | 41.1 | 6.9  | 22.7 | 41.8 | 12.3 |
> | SFT (100 steps) + RL. (De) | 82.1 | 73.5 | 74.3 | 74.7 | 73.8 | 58.8 | 74.2 | 62.3 | 12.5 | 47.3 | 63.4 | 47.7 |
>
> Results indicate that while the cold-start strategy is effective, better than SFT (Only SFT), directly applying RL (Only RL) generally yields performance superior to or comparable with SFT+RL. For instance, on MGSM, the average score for SFT+RL (De) is 52.6, whereas RL (De) achieves 71.5. This suggests that SFT might cause the model to converge prematurely to specific language patterns or local optima, thereby limiting RL's capacity to explore better strategies for multilingual reasoning. In the revised paper, we have added this comparison analysis in Section 3.2 and "Cold-Start" experimental results in Appendix A.4, Table 15.

---

> ### Author Response · Authors · 2025-11-21
>
> ### **About Transferability Differences Across Algorithms** ###
>
> We deeply appreciate the insightful question regarding why transferability differs between SFT and RL (e.g., Chinese data performs better in SFT, while German data performs better in RL on French questions). In fact, this phenomenon is not observed solely in French questions but remains consistent across various languages questions. Specifically, as shown in Table 1 in the paper, in RL training, the model trained on German data outperforms those trained on Chinese data on the average score (71.5 for De V.S. 66.0 for Zh). Conversely, in SFT, the model trained on Chinese data performs slightly better than its German-trained on the average score (46.9 for Zh V.S. 46.3 for De). This trend is further corroborated by the results on the MMath500 benchmark presented in Table 2 in the paper.
>
> Prior studies mentioned that Perplexity (PPL) plays a critical role in Reinforcement Learning (RL) [1,2]. To investigate the underlying causes of the superior performance of German data in RL, we employed the base model to generate six responses for each question across different languages via sampling. We then calculated the Perplexity (PPL) of these responses. Additionally, we computed the BLEU score between the multiple sampled responses for each question to serve as a metric for Self-Similarity. The results (also shown in Appendix A.4, Table 11) are:
>
> |                 | En    | Zh    | De    | Fr    | Ja    | Es    |
> | --------------- | ----- | ----- | ----- | ----- | ----- | ----- |
> | Perplexity      | 1.186 | 1.248 | 1.414 | 1.332 | 1.440 | 1.267 |
> | Self-Similarity | 0.621 | 0.505 | 0.425 | 0.448 | 0.433 | 0.534 |
>
> We found that German questions exhibit high PPL (1.414) and the lowest Self-Similarity (0.425 for De vs. 0.621 for En). This indicates that the model faces higher "uncertainty" when processing German questions. In the exploration phase of RL, this uncertainty potentially prompts the model to step out of single-language constraints and attempt reasoning using mixed languages or its strongest language (English), thereby inadvertently activating stronger cross-lingual generalization capabilities. In contrast, SFT tends to closely fit the language distribution of the training data, thus relying mainly on superficial similarities between languages for transfer.
>
> Furthermore, in addition to German, both French and Japanese questions elicit relatively high PPL values and low Self-Similarity scores. This aligns with the results reported in Table 7 in the Appendix A.4 in the paper, which demonstrate that using Japanese and French data for RL training yields superior performance and generalization capabilities. In the revised paper, we have added this analysis regarding Perplexity and Self-Similarity to Section 4.2 and Table 11 in Appendix A.4.
>
> We sincerely thank you for this valuable suggestion, which has guided us to uncover underlying factors previously overlooked. This finding prompts further in-depth investigation and provides a promising direction for our future work in enhancing multilingual capabilities.
>
> **References**
>
> [1] Perplexity-aware correction for robust alignment with noisy preferences.
>
> [2] Rectify Evaluation Preference: Improving LLMs' Critique on Math Reasoning via Perplexity-aware Reinforcement Learning.

---

> ### Author Response · Authors · 2025-11-21
>
> ### **About Mixed-Language Training** ###
>
> We appreciate the suggestion to evaluate the performance of training on combined languages, which is a standard setting for multilingual models. We added an experiment training on a mixture of English, Chinese, and German datasets (Mix). To ensure a fair comparison, the total data volume of the mixed dataset is consistent with that of the single-language training. The results (also shown in Appendix A.4, Table 14 ) are:
>
> | Models    | En   | Zh   | De   | Es   | Fr   | Ja   | Ru   | Th   | Sw   | Bn   | Avg  | Gen  |
> | --------- | ---- | ---- | ---- | ---- | ---- | ---- | ---- | ---- | ---- | ---- | ---- | ---- |
> | Base      | 63.4 | 48.3 | 33.5 | 57.7 | 38.9 | 19.5 | 30.3 | 17.6 | 7.3  | 1.2  | 31.8 | 0.0  |
> | SFT (En)  | 64.7 | 54.5 | 50.7 | 56.4 | 56.2 | 36.9 | 55.5 | 44.1 | 6.9  | 26.2 | 45.2 | 18.1 |
> | RL (En)   | 85.8 | 72.1 | 70.8 | 77.3 | 76.6 | 61.2 | 64.9 | 61.0 | 9.5  | 47.5 | 62.7 | 49.1 |
> | SFT (Zh)  | 65.7 | 58.7 | 48.4 | 55.7 | 56.1 | 43.5 | 56.6 | 45.8 | 7.5  | 30.5 | 46.9 | 20.4 |
> | RL (Zh)   | 86.1 | 76.3 | 74.2 | 81.1 | 76.1 | 64.5 | 78.1 | 64.9 | 10.3 | 48.3 | 66.0 | 52.6 |
> | SFT (De)  | 63.9 | 54.2 | 57.5 | 55.7 | 52.8 | 39.3 | 55.1 | 47.6 | 8.4  | 28.8 | 46.3 | 19.3 |
> | RL (De)   | 91.0 | 77.6 | 80.5 | 82.7 | 80.0 | 67.8 | 81.3 | 75.3 | 15.9 | 63.3 | 71.5 | 60.4 |
> | SFT (Mix) | 65.3 | 55.9 | 52.9 | 56.3 | 53.0 | 42.9 | 53.2 | 47.1 | 7.6  | 29.8 | 46.4 | 19.6 |
> | RL (Mix)  | 87.9 | 75.4 | 77.1 | 79.0 | 79.3 | 64.1 | 78.2 | 69.7 | 12.6 | 57.7 | 68.1 | 55.2 |
>
> Results show that while RL (Mix) achieves very competitive results (Average 68.1), outperforming RL (En) and RL (Zh), RL (De) (trained on German only) still maintains the highest generalization performance (Average 71.5). This further corroborates the core finding of our paper: some specific non-English languages (with high difficulty/low self-similarity) can stimulate the model's generalization potential in RL training more effectively than even mixing multiple languages. In the revised paper, we have added this comparison analysis of mixed-data in Section 3.3 and the results in Appendix A.4, Table 14.
>
> ### **About Language Consistency Reward Definition** ###
>
> We sincerely appreciate the suggestion for clarification on the Language Consistency Reward. In Equation 1, the Language Consistency Reward is designed to explicitly encourage the model's output language to match the language of the input instruction. For implementation,  we use langid as a language identifier to detect the primary language of the generated response. The verifier checks if the detected language matches the language of the prompt. A positive reward is given for a match, and a penalty is applied otherwise. We have added a detailed description of this implementation to the revised paper in Section 4.1.
>
> Thank you again for your valuable feedback. We are particularly grateful for your insightful comments regarding the transferability differences, which highlighted the importance of our exploration into non-English RL. Your suggestions have helped us improve the clarity and comprehensiveness of our work.

---

### Author Response · Authors · 2025-11-29
**Summary of Rebuttal Improvements**

We sincerely thank all reviewers for their constructive feedback. During the rebuttal, we are particularly encouraged that the responding reviewers have acknowledged our efforts in addressing their concerns through detailed explanations and additional experiments (**Reviewer xepT raised the Rating from 4 to 6 on November 23rd, Reviewer 58H9 raised the Soundness score on November 26th**).

Across reviews, there is **broad agreement on the strengths of our work**: the systematic evaluation demonstrating RL's superiority over SFT in cross-lingual generalization, and most notably, the counter-intuitive discovery that RL training on non-English data yields superior generalization compared to English data. Reviewers further highlighted the solidity of our experiments and the insightful analysis of underlying factors, which serves as a crucial guide for multilingual reasoning.

___

During the rebuttal, we engaged extensively with reviewers to address their concerns through comprehensive discussions and experimental additions:

**Scale Extension and Task Expansion (Reviewers fFzm, xepT):** Reviewers raised concerns about the limitation to  model's scale and task scale. To further validate robustness, we conducted scale extension experiments on Qwen2.5-7B-Base and task expansion on instruction following (M-ifEval) and logical reasoning (Multilingual LogiQA). Consistent gains across scales and domains proved the generalizability of our findings, addressing concerns regarding model scale and domain coverage.

**Mechanistic Analysis and Causal Verification (Reviewers hEuo, fFzm, 58H9):** Regarding German Data's Superiority Mechanism, we analyzed Perplexity and Self-Similarity, finding that German data exhibits higher perplexity and lower self-similarity. This high "uncertainty" forces the model to engage in greater sampling diversity during RL exploration, activating stronger cross-lingual capabilities.  This totally inspired us to delve deeper into the mechanisms behind the German data's superiority in RL. Regarding Language Usage, our Rank Score analysis revealed that RL models (especially RL-De) significantly increase the use of mixed languages (notably English) for reasoning, demonstrating an adaptive language selection strategy driven by uncertainty. Furthermore, through Causality Verification using "Inconsistency Prompt/Reward" experiments, we confirmed that while language inconsistency contributes to performance, the model's autonomous exploration (standard RL) yields the optimal results.

**Baselines Supplementation and Mixed Language Training (Reviewers hEuo, 58H9):** Reviewers suggested comparing with stronger practical baselines and verifying inference settings. To validate practical effectiveness, we performed baseline supplementation (Cold-Start SFT+RL and Few-Shot inference) and mixed training experiments (En+Zh+De). Results showing that direct RL generally outperforms SFT+RL, indicating that SFT may cause premature convergence to local optima. Meanwhile, consistent trends in few-shot settings and the superiority of specific non-English languages over mixed training confirmed that specific languages effectively stimulate generalization potential, addressing the concern about experimental completeness.

**Data Validity and Implementation Details (Reviewers xepT, 58H9):** Reviewers questioned translation quality and specific settings. We validated data reliability through head-to-head comparisons with MGSM8K-Instruct and provided full implementation details (RFT, Inference settings, and Training Settings). High win rates and reproducibility checks addressed reviewers' concerns about data quality and rigor.

Through **10 major improvements**, including **5 types of supplementary experiments** (scale extension, task expansion, cold-start, mixed training, few-shot inference) and **3 categories of in-depth analyses** (German data's superiority mechanism, language usage, causal verification), we systematically responded all major concerns. We sincerely thank all reviewers for their insightful suggestions which significantly strengthened our work!

---

### Meta-Review · Area_Chair_noAo · 2025-12-25

**Summary:**

This paper systematically compares reinforcement learning (RL) and supervised fine-tuning (SFT) for cross-lingual reasoning generalization in language models. Mechanistic analyses suggest that RL promotes more robust and transferable reasoning behaviors, helping preserve pretraining features and reducing semantic drift across languages.

Reviewers gave a mixture of ratings, together with some major comments: Scalability over larger models or datasets (fFzm, xepT, 58H9), limited reasoning tasks (fFzm, xepT), missing implementation or experimental setup details (hEuo, xepT, 58H9), etc. The authors provided detailed responses including more clarifications and extended evaluations. AC has read through all of the reviews and responses, and think the critical concerns have been properly addressed. One reviewer who gave the rating of 4 confirmed to raise the score. Therefore, this paper is recommended for acceptance.

**Reviewer Concerns:**

The major concerns raised by the reviewers should have been addressed by the responses, including:

* Lack of evaluation over larger models and datasets (fFzm, xepT, 58H9)
* Lack of more types of reasoning tasks (xepT, fFzm)
* Missing experimental setup details (xepT, 58H9)
* Missing baselines or more settings (hEuo)
* Missing analysis about transferability (hEuo), causal proof (fFzm), language usage (58H9), translation quality (xepT)

**Reviewer Scores:**

One reviewer xepT confirmed to raise the score from 4 to higher. Another reviewer (fFzm) who gave the rating of 4 has the chance to increase the score if fully involved in the discussion.

---

### Decision · Program_Chairs · 2026-01-26

Accept (Poster)